# FRUGAL: Memory-Efficient Optimization by Reducing State Overhead for Scalable Training

Philip Zmushko [1 2]   Aleksandr Beznosikov [2 3 4 1]   Martin Takáč [5]   Samuel Horváth [5]

## Abstract

With the increase in the number of parameters in large language models, the training process increasingly demands larger volumes of GPU memory. A significant portion of this memory is typically consumed by the optimizer state. To overcome this challenge, recent approaches such as low-rank adaptation (LoRA), low-rank gradient projection (GaLore), and blockwise optimization (BAdam) have been proposed. However, in all these algorithms, the *effective rank of the weight updates remains low-rank*, which can lead to a substantial loss of information from the gradient. This loss can be critically important, especially during the pre-training stage. In this paper, we introduce FRUGAL (**F**ull-**R**ank **U**pdates with **G**r**A**dient sp**L**itting), a new memory-efficient optimization framework. FRUGAL leverages gradient splitting to perform low-dimensional updates using advanced algorithms (such as Adam), while updates along the remaining directions are executed via state-free methods like SGD or signSGD. Our framework can be integrated with various low-rank update selection techniques, including GaLore and BAdam. We provide theoretical convergence guarantees for our framework when using SGDM for low-dimensional updates and SGD for state-free updates. Additionally, our method consistently outperforms concurrent approaches, achieving state-of-the-art results in pre-training and fine-tuning tasks while balancing memory efficiency and performance metrics.

## 1. Introduction

In recent years, Large Language Models (LLMs) such as GPT (OpenAI, 2023) and LLaMA-3 (Dubey et al., 2024) have demonstrated remarkable performance across various disciplines (Brown, 2020; Yang et al., 2024; Romera-Paredes et al., 2024). However, a critical factor in achieving these results is the size of these models (Hoffmann et al., 2022). Increasing the number of parameters leads to higher computational and memory costs. For example, an 8 billion parameter LLaMA-3 model in 16-bit format requires 32GB just for parameters and gradients. Using the standard Adam optimizer (Kingma, 2014) adds another 32GB for $m$ and $v$ statistics. Moreover, achieving high-quality results often requires 32-bit precision for weights and optimizer states (Zamirai et al., 2020), pushing memory requirements beyond even high-end GPUs like the A100-80GB.

Numerous research projects have been aimed at reducing these significant costs. These approaches include engineering solutions like gradient checkpointing (Chen et al., 2016) and memory offloading (Rajbhandari et al., 2020), which do not change the training trajectory. There are also methods that adjust the training algorithm by decreasing the number of trainable parameters (Frankle & Carbin, 2018; Horváth et al., 2024) or their bit precision (Wortsman et al., 2023), as well as optimizer statistics (Dettmers et al., 2021; Shazeer & Stern, 2018; Zhang et al., 2024c).

Parameter-Efficient Fine-Tuning (PEFT) methods, such as LoRA (Hu et al., 2021) and Dora (Liu et al., 2024b) reduce memory costs by training a relatively small number of parameters compared to the size of the original model, while the remaining modules are frozen. This approach has proven effective for the task of efficient fine-tuning of pre-trained models. However, PEFT methods have a fundamental limitation: parameter updates always lie in a low-dimensional subspace $L$, which prevents the use of these methods for pre-training (Lialin et al., 2023) and may restrict their capabilities in fine-tuning (Zhang et al., 2024a).

Recent works, such as GaLore (Zhao et al., 2024a), ReLoRA (Lialin et al., 2023) and BAdam (Luo et al., 2024) offer a solution to this problem. These methods enable higher-dimensional full-parameter learning by periodically

---
[1]Yandex, Russia [2]Moscow Institute of Physics and Technology, Russia [3]Ivannikov Institute for System Programming RAS, Russia [4]Skolkovo Institute of Science and Technology, Russia [5]Mohamed bin Zayed University of Artificial Intelligence, UAE. Correspondence to: Philip Zmushko <zmushko.ph.a@gmail.com>.

*Proceedings of the 42nd International Conference on Machine Learning*, Vancouver, Canada. PMLR 267, 2025. Copyright 2025 by the author(s).

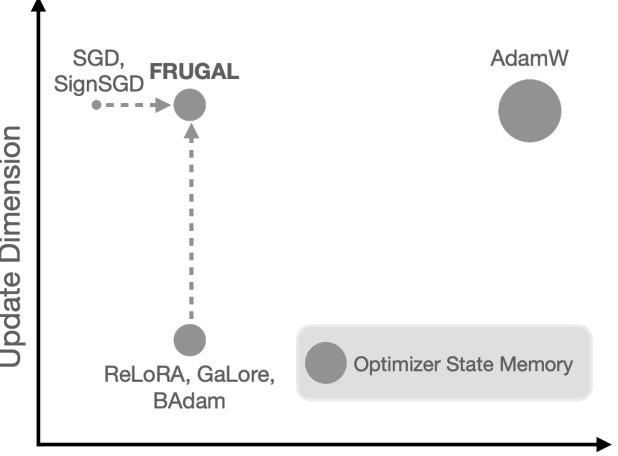

*Figure 1.* FRUGAL reduces memory usage by splitting gradient updates into low-dimensional updates with advanced optimizers (e.g., AdamW) and using state-free methods (e.g., SignSGD and SGD) for the rest.

changing the optimizable low-rank subspace $L$. However, even though these methods result in overall parameter changes that are high-dimensional, the updates in each step remain low-dimensional. The dimensionality of the frozen subspace $\dim M = \dim L^\perp$ significantly exceeds $\dim L$. The remaining information contained in the gradient is not utilized for parameter updates. Nevertheless, this information can still be leveraged to train the model.

We present the FRUGAL framework, designed to bridge this gap. Our approach stems from a crucial observation: although memory constraints prevent using optimizers with auxiliary optimizer state — such as Adam (Kingma, 2014) — in the remaining subspace $M$, one still can update $M$ using state-free optimization algorithms like Stochastic Gradient Descent (SGD) or signSGD (Bernstein et al., 2018). This solution allows for high-dimensional updates, which provides additional opportunities to explore the parameter space and improves convergence. We will further refer to the subspaces $L$ and $M$ according to the types of optimizers used for their updates - **state-full** and **state-free**.

**Contributions.** We summarize the main contributions of our work as follows:

- We present a new memory-efficient optimization framework that combines the use of advanced optimization algorithms for the state-full subspace with state-free algorithms for the complementary subspace. The framework supports various types of state-full optimizers, state-free optimizers, and different methods for projecting the gradient onto the state-full subspace.

- We provide theoretical convergence guarantees for our framework. In the proof, we consider the case with SGDM as the state-full optimizer and SGD as the state-

---

**Algorithm 1** FRUGAL (State-Full, State-Free)

**Input:** model $f_\theta$ with $p$ parameter sets $\{\theta_i \in \mathbb{R}^{d_i}\}_{i=1}^p$, loss $\mathcal{L}$, gradient projectors $\{P_{k,i}\}_{i=1}^p$, number of steps $K$

1: **for** $k = 1, 2, \ldots K$ **do**
2:      get data batch $(x, y)$
3:      compute $\ell \leftarrow \mathcal{L}(f_\theta(x), y)$ {Forward}
4:      **for** $g_i = \frac{\partial \ell}{\partial \theta_i}$ from Backward **do**
5:          $g_{\text{full},i} \leftarrow P_{k,i}(g_i)$, {Project Grad}
6:          $g_{\text{free, i}} \leftarrow g_i - P_{k,i}^{-1}(g_{\text{full},i})$ {Residual}
7:          $s_{\theta_i} \leftarrow [P_{k,i}(P_{k-1,i}^{-1}(s), \; s \in s_{\theta_i}]$ {Project state}
8:          $u_{\text{full, i}} \leftarrow \texttt{State-Full.update}(\theta_i, g_{\text{full},i}, s_{\theta_i})$
9:          $u_{\text{free, i}} \leftarrow \texttt{State-Free.update}(\theta_i, g_{\text{free},i})$
10:         $\theta_i \leftarrow \theta_i + P_{k,i}^{-1}(u_{\text{full},i}) + u_{\text{free, i}}$
11:      **end for**
12: **end for**

---

free optimizer, and we show that FRUGAL matches the best-known convergence rate in many scenarios.

- To verify the practical applicability of FRUGAL, we conduct extensive experiments in popular real-world scenarios[1]. In these experiments, we pre-train LLaMA-like models (up to 1B parameters) on the Colossal Clean Crawled Corpus (C4) dataset (Raffel et al., 2020) and fine-tune RoBERTa (Liu, 2019) on the GLUE benchmark (Wang, 2018). The results show that our method significantly outperforms previous memory-efficient algorithms while using less memory budget.

- We demonstrate that only the Output layer in transformer-like models requires advanced optimizers like Adam, while other modules (including RMSNorms and Embeddings) can use simpler methods like signSGD without significant performance loss. This opens up new possibilities for memory-efficient training and provides crucial insights into Transformers learning dynamics.

## 2. Related work

**Memory-efficient full-parameter learning.** Recent research has focused on reducing the memory footprint of LLM by decreasing the size of the optimizer states while maintaining their performance. Low-rank adaptation methods, such as LoRA (Hu et al., 2021), inject trainable rank decomposition matrices into linear layers, reducing memory requirements by optimizing only a few learnable adapters. ReLora (Lialin et al., 2023) builds upon this by merging low-rank adaptations into the main model weights during training, increasing the total rank of the update. BAdam (Luo et al., 2024) leverages Block Coordinate Descent for full-parameter training by switching active blocks during fine-

---

[1]The code is available at https://anonymous.4open.science/r/FRUGAL-D3CA.

tuning. MicroAdam (Modoranu et al., 2024) compresses gradient information before feeding it into the optimizer state, significantly reducing the memory footprint while enabling full parameter learning with error feedback mechanisms. GaLore (Zhao et al., 2024a) maintains full parameter learning by projecting gradients onto a low-rank subspace using SVD decomposition, storing optimizer states in this reduced space. However, while these methods effectively reduce memory overhead, they all perform *low-rank updates at each iteration*. In contrast, our approach utilizes all available gradient information to perform *full-dimensional updates at each optimizer step*, offering a novel perspective on memory-efficient optimization for LLM.

However, we note that there also exist several concurrent works — Fira (Chen et al., 2024a), LDAdam (Robert et al., 2024), and Adamem (Vyas et al., 2024) — that also adopt full-dimensional updates for similar goals. See Appendix B for a comparison and detailed discussion.

**Other memory-efficient optimization.** Several other methods have been proposed to reduce the memory footprint of optimizers. AdaFactor (Shazeer & Stern, 2018) attempts to mimic Adam's behavior while reducing memory usage through factorization of the variance matrix $v$. Adam-mini (Zhang et al., 2024c) further reduces memory by storing only one value $v$ per block. Dettmers et al. (2021) and Li et al. (2024) decrease memory footprint by quantizing optimizer states to lower-precision representations. Lv et al. (2023) proposed to reduce memory by fusing the backward operation with the optimizer update. Notably, these approaches are *orthogonal* to our method FRUGAL and *can be combined with it* for further memory efficiency.

**Block Coordinate Descent.** Block Coordinate Descent (BCD) is a well-established optimization method with a rich history in mathematical optimization (Ortega & Rheinboldt, 2000; Tseng, 2001; Richtárik & Takáč, 2014; 2015a;b). In recent years, a specific instance of BCD, known as *layer-wise learning,* has been applied to deep learning. Notable examples include Luo et al. (2024); Pan et al. (2024), which leverage this approach for LLM fine-tuning. To the best of our knowledge, our work presents **the first theoretical analysis** of an extended BCD framework (Section 5) where the *remaining layers are also updated with a different algorithm*. This novel approach extends traditional BCD techniques, opening new avenues for full model optimization.

**Sign-based methods for training language models.** Since its introduction, Adam has become the de facto primary optimization algorithm, demonstrating superior practical results compared to SGD-based algorithms across various deep learning tasks. This difference is particularly noticeable when training Transformers on language tasks. While Zhang et al. (2020) hypothesized that Adam outperforms SGD in this setup due to *the heavy-tailed distribution of sampling-*

*Table 1.* Comparison of different projection and state-free subspace optimization strategies on pre-training LLaMA-130M on C4 with AdamW as the state-full algorithm.

| Projection type | Optimizes state-free subspace | Validation perplexity ↓ | | |
|---|---|---|---|---|
| | | 4k | 40k | 200k |
| SVD | No | **39.75** | 24.38 | 21.11 |
| Random | No | 42.31 | **23.55** | **20.01** |
| Random | Yes | 37.26 | 21.53 | 18.64 |
| SVD | Yes | **33.96** | **21.01** | **18.35** |
| RandK | Yes | 36.38 | 21.25 | 18.63 |
| Blockwise | Yes | 37.20 | 21.42 | 18.60 |
| AdamW | | 33.95 | 20.56 | 18.13 |

*induced errors,* Kunstner et al. (2023) demonstrated that this superiority persists even in full-batch training. They proposed a new hypothesis suggesting that Adam's key success factor is related to *its similarity to signSGD* (Balles & Hennig, 2018; Balles et al., 2020), and both Kunstner et al. (2023) and Zhao et al. (2024b) showed that the signed descent with momentum reduces the performance gap with Adam. In contrast, to the best of our knowledge, **we are the first to train the majority of language model parameters using signSGD without momentum**, achieving minimal loss in quality. This approach further demonstrates the effectiveness of sign-based methods for LLM training, paving the way for more efficient and scalable optimization strategies.

## 3. Empirical Analysis and Motivation

### 3.1. The importance of exploring the entire space during the training process

In recent work, Zhao et al. (2024a) proposed GaLore, an optimization method based on projecting the gradient matrix $G$ of each Linear layer[2] onto a low-dimensional subspace. To obtain the projection matrix $P$, they use the SVD decomposition of $G_t$, which is recomputed with frequency $T$. The vectors or rows of $G$ are projected onto the first $r$ left or right singular vectors, respectively. This approach has theoretical foundations: the first $r$ singular vectors correspond to the first $r$ singular values and, therefore, should better utilize information from the spectrum of $G$.

Given the computational burden of SVD decomposition, a natural question arises about the possibility of employing a random semi-orthogonal projection matrix $R$ as an alternative to projecting onto the first $r$ singular columns with $P$. Surprisingly, while the SVD decomposition provides better initial performance, the random projection proves superiority in long-term training, yielding significant improvements. As an illustration, we took the pre-training[3]

---

[2]Since Linear layers contain most parameters and require most memory, we primarily focus on them.

[3]See Section 6.1 for a detailed description and discussion.

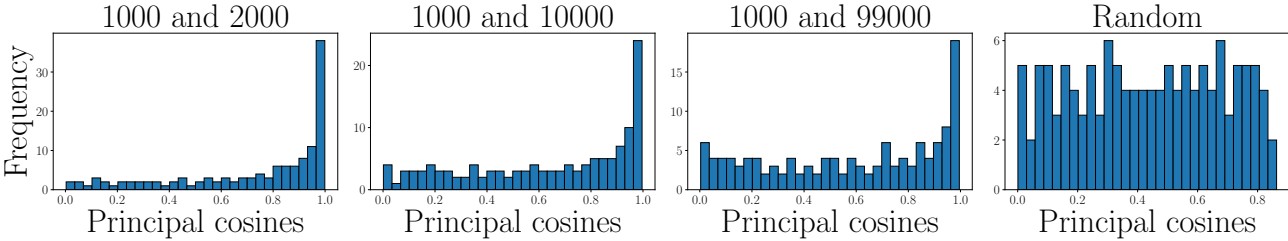

*Figure 2.* Histograms of principal angle cosines. The first three are taken between $P_t$ and $P_{t'}$ from different iterations $t$ and $t'$. $P$ is obtained from the truncated SVD decomposition of the gradient $G$ of the Key projection from the 5th layer. The last histogram is taken between two random semi-orthogonal projections $R$ and $R'$ for comparison.

of a 130M model with LLaMA-like architecture on the C4 dataset. The results are presented in the first part of Table 1, where we compare SVD and Random projections.

To investigate this phenomenon, we pre-trained the LLaMA-60M model and collected gradients $G_t$ from different iterations $t$ for examination. We evaluated the similarity of the projection matrices by calculating the principal angles between the projections $P_t$ from different steps. Similarly to the observations in Q-Galore (Zhang et al., 2024d), we found that these projections show minimal change during training; see Figure 2 for details.

Here, we take the projection matrix of k_proj from 5-th layer and plot histograms of the cosine of the principal angles between pairs $P_t$ and $P_{t'}$ from different iterations. For comparison, we also include the random projections on the right. As can be seen, the distributions of cosines differ significantly for $P_t$ and for $R_t$. While $R_t$ feature no angles with cosines higher than 0.9, the top 57 cosines for $P_t$ surpass 0.9, even for gradients 1000 steps apart.

This leads to the conclusion that although the SVD decomposition generally better captures the information contained in $G_t$, the original GaLore algorithm updates the weights only in a small subspace. We hypothesize that training with random projections yields superior results due to the more extensive investigation of the optimizable space during the training process. *This finding indicates that to achieve better convergence, it is important to find optimization algorithms that explore the entire space during the training process.*

### 3.2. Advantage of the Full-Rank Updates

The insight from Section 3.1 suggests that the training of language models performs significantly better when the entire parameter space is explored during the training process. Given the importance of updating parameters in all directions, this poses the question: *Is it optimal to use low-rank updates, as employed by methods such as GaLore, ReLoRA, and BAdam?* The effective rank of low-rank updates is significantly smaller than the full dimensionality of the parameter space, inevitably leading to a loss of valuable information contained in the gradient.

However, the method to leverage the full-rank gradient for updating parameters is not readily obvious. Using algorithms like Adam (Kingma, 2014) is not an option due to the memory overhead they introduce, which is exactly what we aim to avoid. An alternative approach is to use state-free optimizers such as SGD or signSGD (Bernstein et al., 2018). Unfortunately, SGD has been shown to be ineffective for training transformer models, as shown in Zhang et al. (2020); Pan & Li (2023).

Nevertheless, a recent study Zhao et al. (2024b) suggests a promising methodology: while SGDM generally does not work well with transformers, using SGDM for the majority of parameters and Adam for a selected subset can lead to effective training. This raises the question: Could a hybrid approach using SGD or signSGD instead of SGDM be viable? If the key subset of parameters is handled by advanced algorithms, can the other parameters be trained effectively with state-free optimizers?

To address this question, we conducted an experiment on LLaMA-130M, where we utilized the Adam (Kingma, 2014) for state-full parameters and signSGD (Bernstein et al., 2018) for state-free parameters[4]. Once again we used Random projection and highlighted the result in the second part of Table 1. Full-rank updates significantly enhance performance, approaching the efficiency of the memory-intensive Adam optimizer. *These findings underscore the potential of state-free algorithms for updating a substantial portion of the parameter space, paving the way for efficient and scalable optimization methods that deliver high performance without the significant memory costs traditionally associated with state-of-the-art optimizers.*

## 4. Full-Rank Updates with GrAdient spLitting

**General framework.** The setup outlined in the conclusion of Section 3.2 results in a general framework for memory-efficient optimization. It operates as follows: the entire space is partitioned into *state-full* and *state-free* subspaces. The state-full subspace is updated using an advanced algorithm, while the state-free subspace is updated using a state-free method. After a certain number of steps, the state-

---

[4]See detailed description of the setup in Appendix A.1

full subspace is changed to better explore the optimization space. A formal description is presented in Algorithm 1.

We note that this framework allows for variation not only in the *state-full* optimizer but also in the choice of *projection* and *state-free* optimizer. However, determining the optimal state-free optimizer and the projection method onto the state-full subspace is not readily apparent. In this section, we strive to find the optimal configuration.

**State-free optimizer.** We conducted a preliminary experiment using different state-free algorithms to choose between SGD and signSGD (Bernstein et al., 2018). Table 10 shows that signSGD outperforms SGD, leading us to favor signSGD. We attribute this performance to the similarities between signSGD and Adam (Kingma, 2014), as noted in Balles & Hennig (2018); Balles et al. (2020); Kunstner et al. (2023). Additionally, signSGD produces updates of similar magnitude to those generated by Adam, which simplifies the calibration of the learning rate for state-free parameters.

**Projection type.** When selecting a projection method, it is crucial to strike a balance between quality and memory efficiency. When using SVD decomposition for projection matrices, as in GaLore (Zhao et al., 2024a), the method better preserves the information embedded in the gradient but requires additional memory for storing projection matrices and computational resources for performing the SVD. To reduce computational demands, one could employ random coordinate projection denoted as RandK, but this requires additional memory or recomputation[5]. A more structured alternative is to select not random entries but entire random columns or rows. The most aggressive approach follows the method from BAdam, wherein an entire block is chosen as the state-full subspace. The performance results obtained with all these variants are presented in the second part of Table 1. SVD slightly outperforms both RandK and Block projections, demonstrating comparable performance. Nonetheless, a downside is the increased compute and memory demand from SVD. Therefore, we opt for the blockwise selection, as it is the most memory-efficient — requiring only the storage of active block indices.

In experiments in Section 6, we use a specific variant with AdamW as the State-Full optimizer and signSGD as the State-Free optimizer. We primarily employ blockwise projection but switch to column-wise projection when the number of parameters in any single block exceeds memory budget, as detailed in Section 7. In addition, PyTorch-like pseudocode of our framework is presented in Appendix G.

For Line 7, state projection, in Algorithm 1, we note that if the projection does not change, i.e., $P_{k,i} = P_{k-1,i}$, then $P_{k,i}(P_{k-1,i}^{-1}(s)) = s$. Thus, we only need to project states

---

[5]See Appendix C for discussion on the memory requirements for different projection methods.

---

**Algorithm 2** FRUGAL (SGDM, SGD)

**Input:** momentum weight $\beta \in [0, 1)$, initialization $x^1 \in \mathbb{R}^d$ and $m^0 = 0$, step sizes $\{\alpha^k > 0\}_{k=1}^K$, momentum set $J_k \subset [d]$ for $k = 1, 2, \dots$

1: **for** $k = 1, 2, \dots$ **do**
2: $\quad \tilde{g}^k \leftarrow \nabla f_{\zeta^k}(x^k)$
3: $\quad \tilde{m}_j^k \leftarrow (1-\beta)\tilde{g}_j^k + \beta \begin{cases} \tilde{m}_j^{k-1} & \text{if } j \in J_k, \\ 0 & \text{otherwise;} \end{cases}$
4: $\quad \tilde{u}_j^k \leftarrow \begin{cases} \tilde{m}_j^k & \text{if } j \in J_k, \\ \tilde{g}_j^k & \text{otherwise;} \end{cases}$
5: $\quad x^{k+1} \leftarrow x^k - \alpha^k \tilde{u}^k$
6: **end for**

---

when the projection changes from one round to another. However, our preliminary experiments with RandK selection showed that resetting states performs comparably to projection. Therefore, we could replace this projection with state resetting when the projection changes, which also aligns with blockwise subspace selection. However, either resetting or projecting states is important since we want projected gradients and optimizer states to reside in the same space. For instance, GaLore ignores this step, which leads to degraded performance when projections are updated frequently; see Appendix D and Section 6.4 for details.

## 5. Theoretical Results

For the theoretical analysis, we consider the case where the *State-Free* optimizer is SGD and the *State-Full* optimizer is SGD with momentum (SGDM). For the projection, we use coordinate-wise projection. This special case of FRUGAL is provided in Algorithm 2. We minimize the objective

$$\min_{x \in \mathbb{R}^d} \left\{ f(x) := \mathbb{E}_{\zeta^k}[f_{\zeta^k}(x)] \right\}, \tag{1}$$

where we access $f$ via a stochastic oracle that takes $x$ as input and returns $(f_{\zeta^k}(x), \nabla f_{\zeta^k}(x))$.

### 5.1. Notation and Preliminaries

We use $\|\cdot\|$ for the vector $\ell_2$-norm, and $\langle \cdot, \cdot \rangle$ stands for the dot product. Let $g^k$ denote the full gradient of $f$ at $x^k$, i.e., $g^k := \nabla f(x^k)$, $\tilde{g}^k$ denote the stochastic gradient $\tilde{g}^k = \nabla f_{\zeta^k}(x^k)$ for random sample $\zeta^k$, and $f^* := \min_{x \in \mathbb{R}^d} f(x)$. We use subscript $j$ to denote the $j$-th coordinate. We call a function L-smooth if it is continuously differentiable and its gradient is Lipschitz continuous:

$$\|\nabla f(x) - \nabla f(y)\| \le L\|x - y\|. \tag{2}$$

**Assumption 5.1.** We make the following assumptions, which are standard in non-convex stochastic optimization; see (Liu et al., 2020).

*Table 2.* Comparison of validation perplexity and memory estimation for various optimization methods across LLaMA model scales trained on C4. We also indicate the additional memory overhead introduced by the optimization algorithm. The values are calculated assuming that each float value occupies 4 bytes (float32). $\rho$ denotes the proportion of the Linear layer parameters in the state-full subspace. Note that Embeddings, RMSNorms, and Output layer are always trained with AdamW.

|  | 60M | 130M | 350M | 1B |
|---|---|---|---|---|
| AdamW | 22.73 (0.43G) | 18.13 (1.00G) | 14.43 (2.74G) | 12.02 (9.98G) |
| GaLore, $\rho = 0.25$ | 25.68 (0.30G) | 21.11 (0.54G) | 16.88 (1.10G) | 13.69 (3.41G) |
| BAdam, $\rho = 0.25$ | 24.86 (0.29G) | 20.34 (0.52G) | 16.41 (1.05G) | 13.75 (3.23G) |
| FRUGAL, $\rho = 0.25$ | **23.59** (0.29G) | **18.60** (0.52G) | **14.79** (1.05G) | **12.32** (3.23G) |
| FRUGAL, $\rho = 0.0$ | 24.06 (0.24G) | 18.90 (0.37G) | 15.03 (0.49G) | 12.63 (0.98G) |
| Training tokens | 20B | 20B | 24B | 30B |
| Number of iterations | 200k | 200k | 240k | 300k |

1. **Smoothness:** The objective $f(x)$ in equation 1 is $L$-smooth (Equation (2)).

2. **Unbiasedness:** At each iteration $k$, $\tilde{g}^k$ satisfies $\mathbb{E}_{\zeta^k}[\tilde{g}^k] = g^k$.

3. **Independent samples:** The random samples $\{\zeta^k\}_{k=1}^{\infty}$ are independent.

4. **Bounded variance:** The variance of $\tilde{g}_j^k$ with respect to $\zeta^k$ satisfies $\text{Var}_{\zeta^k}(\tilde{g}_j^k) = \mathbb{E}_{\zeta^k}[\|\tilde{g}_j^k - g_j^k\|^2] \leq \sigma_j^2$ for some $\sigma_j^2 > 0$. We denote $\sigma^2 = \sum_{j=1}^{d} \sigma_j^2$.

Finally, we define the probability that index $j \in J_k$ is selected, conditioned on the prior iteration $k - 1$, as $p_j^k := \text{Pr}_{k-1}[j \in J_k]$. Other useful quantities are $p_{\max}^k := \max_{j \in [d]}\{p_j^k\}$ and $p_{\min}^k := \min_{j \in [d]}\{p_j^k\}$.

### 5.2. Convergence of Algorithm 2

Below, we present the main convergence theorem.

**Theorem 5.2.** *Let Assumption 5.1 hold and $\alpha^k = \alpha \leq \frac{1-\beta}{L(4-\beta+\beta^2)}$. Then, the iterates of Algorithm 2 satisfy*

$$\frac{1}{k}\sum_{i=1}^{k}\mathbb{E}[\|g^i\|^2] = \mathcal{O}\bigg(\frac{f(x^1) - f^*}{k\alpha} + $$
$$+ L\alpha\sigma^2\Big(1 + \frac{\hat{p}_{\max}^k(1 - \bar{p}_{\min}^k)\beta}{(1-\beta)}\Big)\bigg),$$

*where $\bar{p}_{\min}^k = \frac{1}{k}\sum_{i=1}^{k}\bar{p}_{\min}^i$ and $\hat{p}_{\max}^k = \max_{i \in [k]}\{p_{\max}^i\}$.*

The proof is deferred to Appendix E. Let us analyze the obtained result. Firstly, if $J_k = [d]$ or $J_k = \emptyset$, Algorithm 2 becomes SGDM and SGD, respectively. In this case, we have $\bar{p}_{\min}^k = 1$ for SGDM and $\hat{p}_{\max}^k = 0$ for SGD. Therefore, the resulting rate is $\mathcal{O}\left(1/k\alpha + L\alpha\sigma^2\right)$, which recovers the best-known rate for both SGD and SGDM under these assumptions (Liu et al., 2020). Furthermore, if at each step each coordinate is sampled independently with probability $p$, we have $\bar{p}_{\min}^k = \hat{p}_{\max}^k = p$. Therefore, we recover the

*Table 3.* Perplexity and memory consumption (weights, gradients and optimizer states) of different size LLaMA models pre-trained on C4 for 100k iterations (10B tokens) using AdamW with pure bf16 of mixed precision.

| Model size | Format | Memory | Perplexity |
|---|---|---|---|
| 175M | Mixed Precision | 2.0GB | **17.43** |
| 350M | Pure bf16 | 2.1GB | 17.75 |
| 350M | Mixed Precision | 4.2GB | **15.16** |
| 1.3B | Pure bf16 | 7.7GB | 16.51 |

same rate if $p = \mathcal{O}(1 - \beta)$ or $p = \mathcal{O}(\beta)$. Finally, in the worst case (e.g., $J_k$ is deterministic and $0 < |J_k| < d$), we have $\bar{p}_{\min}^k = 0$ and $\hat{p}_{\max}^k = 1$. Thus, the rate becomes $\mathcal{O}\left(1/k\alpha + L\alpha\sigma^2/1-\beta\right)$, which is worse by a factor of $1/1-\beta$. However, this is expected since the bias from momentum is not outweighed by the variance reduction effect, as only the coordinates with momentum enjoy reduced variance; see Lemmas E.2 and E.3 in the appendix for details.

## 6. Pre-training experiments

In this section, we evaluate the performance of FRUGAL on the language models pre-training.

### 6.1. Comparison to existig memory-efficient algorithms

To begin, we compare our framework with existing memory-efficient methods across four sizes of LLaMA-based architectures: 60M, 130M, 350M, and 1B.

**Setup.** The core setup for pre-training is taken from Zhao et al. (2024a). We utilize LLaMA-based (Touvron et al., 2023a) model architectures and train them on the Colossal Clean Crawled Corpus (C4) dataset (Raffel et al., 2020). The C4 dataset is intended for pre-training, making this setup a good approximation of real-world applications. A detailed description of the setup can be found in Appendix A.1.

However, we made several critical modifications compared to Zhao et al. (2024a) to align the experimental setup with

*Table 4.* Perplexity of LLaMA-130M models pre-trained on C4 for 100k iterations (10B tokens). The leftmost column indicates the modules moved to the state-free set and trained using signSGD. The results show that **Output layer**, unlike Embeddings and RM-SNorms, are exceptionally responsive to the choice of optimization algorithm from AdamW to signSGD.

| State-free modules | Perplexity ↓ |
|---|---|
| Linear (FRUGAL $\rho = 0.0$ from Table 2) | 20.02 |
| Linear, *RMSNorms* | 20.07 |
| Linear, *Embeddings* | 20.48 |
| Linear, *Embeddings, RMSNorms* | 20.55 |
| Linear, **Output layer** | 34.66 |

practical training scenarios. Below, we discuss each modification and provide a detailed rationale for these decisions.

- **Training Duration.** The training approach in Zhao et al. (2024a) aligns with the empirical rule from scaling laws (Hoffmann et al., 2022), which suggests using approximately 20 times the size of the model in tokens for training. However, this number of tokens is far from achieving convergence. In practice, models are typically trained for significantly longer periods (Touvron et al., 2023b; Zhang et al., 2024b). One reason for this discrepancy is that the original scaling laws do not account for the inference of the model after training (Sardana & Frankle, 2023). For our experiments, we chose 200k steps for the 60M and 130M models, 240k for the 350M model, and 300k for the 1B and 3B models.

- **Mixed Precision.** Pure 16-bit training has been shown to potentially compromise model convergence and accuracy (Zamirai et al., 2020). This degradation occurs because formats such as float16 or bfloat16, used to store master weights, lack the numerical precision needed for accurate and fine-grained weight updates. Consequently, mixed precision training has become a more common approach for training language models (Le Scao et al., 2023; Almazrouei et al., 2023). Moreover, even when training with fp8, the master weights are typically stored in fp32 format (Liu et al., 2024a).

Our experimental results strongly support the importance of precision choice: in Table 3 we show that adopting pure bf16 training led to such significant performance degradation that doubling the model size failed to compensate for it, effectively negating any memory benefits from reduced precision storage. While training in pure 16-bit format is also possible, stochastic rounding (Gupta et al., 2015; Zamirai et al., 2020) is often employed to mitigate the aforementioned issue. Given that the goal of this research is to identify the optimal optimization algorithm, we deemed it more appropriate to compare optimizers in a transparent and stable setup that does not require auxiliary tricks. Hence, we primarily used Mixed

*Table 5.* Pre-training LLaMA 3B on C4 dataset for 300K steps. Validation perplexity for different iterations is reported.

| Method | 100k | 200k | 300k |
|---|---|---|---|
| AdamW | 14.2 | 12.25 | 10.93 |
| FRUGAL, $\rho = 0.25$ | 14.33 | 12.42 | 11.07 |
| FRUGAL, $\rho = 0.0$ | 14.78 | 12.76 | 11.35 |

Precision training for its illustrative value in understanding each method's potential. However, for completeness, we also conducted experiments in pure bfloat16 format, detailed in our ablation study Section 6.4.

**Baselines.** We use the following methods as baselines:

- **Full-rank Training.** Training using memory-inefficient AdamW (Loshchilov, 2017). Weights, gradients, and statistics are stored and computed for all parameters. This serves as an upper bound for model performance.

- **GaLore.** Zhao et al. (2024a) proposed GaLore, a memory-efficient optimization algorithm that uses a low-rank projection of gradient matrices $G$. Every $T$ steps, the current gradient matrix $G_t$ is used to compute the projection matrix $P$ via SVD decomposition. The gradient is then projected onto the low-rank space, where the optimization step is performed. Subsequently, the resulting low-rank update is projected back into the full-rank space and added to the weights $W$.

- **BAdam.** Luo et al. (2024) proposed a block coordinate descent (BCD)-type optimization method termed BAdam. The parameters are divided into blocks, which are then updated one by one using AdamW. The optimized block is changed every $T$ steps. Although this method was initially proposed only for fine-tuning, it is the closest method to our FRUGAL. Unlike BAdam, in our algorithm, state-free blocks are not frozen but are updated using signSGD.

- **Other Algorithms.** Among other relevant methods, ReLoRA (Lialin et al., 2023), MicroAdam (Modoranu et al., 2024), Fira (Chen et al., 2024a), LDAdam (Robert et al., 2024), and Adamem (Vyas et al., 2024) can also be highlighted. However, we did not include them for comparison here for the following reasons: 1. *ReLoRA*: This method was evaluated in (Zhao et al., 2024a), where it significantly underperformed compared to GaLore. 2. *MicroAdam*: Its current implementation only supports bfloat16 master weights, whereas our main experiments conducted with mixed precision. 3. *Fira, LDAdam, and Adamem:* These methods are concurrent works that were published during the final stages of our work. Accordingly, the majority of our experiments were completed before we became aware of these recent developments.

**Main results.** The results of our experiments are presented in Table 2, which includes both validation perplexity and

*Table 6.* Evaluating FRUGAL for memory-efficient fine-tuning RoBERTa-Base on GLUE benchmark. Results represent the mean and standard deviation across 3 independent runs. Upper ↑ is better.

| Method | Modules | Rank | CoLA | STS-B | MRPC | RTE | SST2 | MNLI | QNLI | QQP | Avg |
|---|---|---|---|---|---|---|---|---|---|---|---|
| Full-parameter | — | — | 63.6 | **91.2** | **90.2** | 78.7 | 94.8 | **87.6** | 92.8 | **91.9** | 86.4 |
| LoRA | QV | 8 | $63.8_{\pm.6}$ | $90.9_{\pm.1}$ | $89.1_{\pm.4}$ | $79.2_{\pm1.1}$ | $94.8_{\pm.2}$ | $87.6_{\pm.2}$ | $93.1_{\pm.1}$ | $90.6_{\pm.0}$ | 86.1 |
| GaLore | All | 8 | $60.0_{\pm.2}$ | $90.8_{\pm.1}$ | $89.0_{\pm.7}$ | $79.7_{\pm.9}$ | $\mathbf{94.9}_{\pm.5}$ | $87.6_{\pm.1}$ | $\mathbf{93.3}_{\pm.1}$ | $91.1_{\pm.1}$ | 85.8 |
| GaLore | QV | 8 | $56.1_{\pm.8}$ | $90.8_{\pm.2}$ | $88.1_{\pm.3}$ | $74.7_{\pm1.9}$ | $94.3_{\pm.1}$ | $86.6_{\pm.1}$ | $92.6_{\pm.1}$ | $89.4_{\pm.1}$ | 84.1 |
| FRUGAL | QV | 8 | $64.5_{\pm.7}$ | $91.1_{\pm.1}$ | $89.2_{\pm.3}$ | $\mathbf{82.4}_{\pm.9}$ | $94.8_{\pm.2}$ | $87.4_{\pm.1}$ | $92.8_{\pm.1}$ | $91.4_{\pm.1}$ | **86.7** |
| FRUGAL | None | 0 | $\mathbf{64.8}_{\pm.5}$ | $91.1_{\pm.1}$ | $89.1_{\pm.3}$ | $81.6_{\pm.6}$ | $\mathbf{94.9}_{\pm.2}$ | $87.3_{\pm.1}$ | $92.8_{\pm.1}$ | $91.3_{\pm.1}$ | 86.6 |

memory footprint estimations for each method. We compared all memory-efficient methods under the same memory budget with a density $\rho = 0.25$. Here, $\rho$ refers to the proportion of Linear layer parameters belonging to the state-full subspace. Similarly to GaLore, non-Linear modules (Embeddings, RMSNorms, Output layer) are optimized with AdamW. See Appendix A.1 for details.

We conducted a grid search to determine the optimal learning rate for AdamW, which we then applied to FRUGAL and BAdam (Luo et al., 2024). For GaLore (Zhao et al., 2024a), we found that using this same learning rate produced better results than the originally suggested rate. This discrepancy might be attributed to our experiments involving a significantly larger number of training steps than those for which GaLore's original learning rate was optimized.

Table 2 demonstrates that FRUGAL significantly outperforms memory-efficient baselines across all model sizes with the same memory budget, coming close to the performance of AdamW.

## 6.2. Zero-density training

Table 2 also reveals a surprising result: FRUGAL with $\rho = 0.0$ outperforms both GaLore and BAdam, even when these competing methods use a higher density of $\rho = 0.25$. Essentially, for FRUGAL with $\rho = 0.0$, the parameters are divided into two parts — a state-full part consisting of the Embeddings, RMSNorms, and Output layer, and a state-free part consisting of all other parameters. This division remains fixed throughout the training. We conducted additional experiments to determine the maximum subset of parameters that can be trained with a state-free optimizer without significant quality degradation. We systematically moved different combinations of the Embeddings, RMSNorms, and Output layer from the state-full to the state-free set and observed the results during the training of LLaMA-130M. Table 4 reveals that the Output layer demonstrates a dramatically higher sensitivity, with changes to its optimizer resulting in severe performance degradation. This finding aligns with results from Zhao et al. (2024b), where the authors demonstrated that most parameters can be trained using SGDM, but the Output layer require training with AdamW.

## 6.3. LLaMA 3B training

To demonstrate the practical viability of our method for large-scale applications, we evaluated FRUGAL against AdamW on the pre-training of the LLaMA 3B model. Due to computational constraints, we conducted a single training run of 300k steps using a cosine learning rate scheduler with 10% warmup steps. We used a learning rate of 5e-4, weight decay of 0.1, and gradient clipping of 1.0, with other hyperparameters consistent with Appendix A.1. The results in Table 5 confirm that FRUGAL successfully scales to billion-parameter models without performance degradation, making it a viable option for industrial-scale applications.

## 6.4. Ablation study

We also conducted additional experiments to verify the robustness of our framework to various hyperparameters.

First, we began by evaluating different model architectures. Experiments with GPT-2 124M (Radford et al., 2019) in Table 12 show that FRUGAL maintains its strong advantage over memory-efficient baselines, albeit with a somewhat wider gap to AdamW. Second, an ablation study on the state-full subspace update frequency $T$ in Table 14 shows that the performance keeps improving up to $T = 200$. We note that, unlike in Zhao et al. (2024a), the perplexity does not decrease significantly even when reducing the update frequency to $T = 10$ ($\sim 0.2$ drop vs. $\sim 4$. drop for GaLore). A detailed explanation for this result can be found in Appendix D. After that, when using other schedulers, the performance gap between FRUGAL and baselines remains consistent, as shown in Tables 15 and 16. Table 8 shows that the same holds for $\beta_2 = 0.95$ — another popular value for the second moment decay parameter in AdamW-like methods. FRUGAL also provides improvement over baselines for other state-full optimizers, as can be seen in experiments with Lion (Chen et al., 2024b) presented in Table 11. Then, the results of the training in pure bfloat16 are presented in Table 9, demonstrating consistency with our main experiments in Table 2, i.e., FRUGAL significantly outperforms the baselines across these variations. We also conducted experiments to show how perplexity changes with varying $\rho$, and the results are presented in Table 17. Finally, we

*Table 7.* Accuracy ↑ comparison of memory-efficient methods on LLaMA 3.1-8B across 8 commonsense reasoning tasks.

| Method | Rank | BoolQ | PIQA | SIQA | HellaSwag | WinoGrande | ARC-e | ARC-c | OBQA | Avg |
|--------|------|-------|------|------|-----------|------------|-------|-------|------|-----|
| LoRA | 32 | 73.39 | 88.19 | 79.48 | 95.13 | 83.50 | 90.24 | 79.52 | 84.00 | 84.18 |
| GaLore | 32 | 74.13 | 88.36 | 78.97 | 94.88 | 83.82 | 91.08 | 80.2 | 85.80 | 84.65 |
| FRUGAL | 32 | **74.25** | **88.47** | 79.12 | **95.15** | 86.11 | **91.37** | 79.86 | 84.80 | 84.89 |
| FRUGAL | 0 | 71.44 | **88.47** | **80.40** | 94.74 | **86.35** | 90.95 | **81.14** | **86.00** | **84.94** |

conducted an experiment to compare different strategies for selecting state-full blocks during training. The results in Table 13 show that there is no significant difference between random and structured block selection.

These experimental results validate that our framework's superiority is resilient to hyperparameter variations.

## 7. Fine-tuning experiments

### 7.1. Fine-tuning RoBERTa on GLUE

We evaluated the performance of our framework in memory-efficient fine-tuning using the GLUE benchmark (Wang, 2018), a widely-used collection of tasks for evaluating language models. Following the approach from Zhao et al. (2024a), we fine-tuned RoBERTa-base (Liu, 2019) using LoRA (Hu et al., 2021) and GaLore as baselines for comparison. We adhered to the setup described in LoRA, where low-rank updates of rank 8 were applied only to the Q and V matrices. See detailed description in Appendix A.2.

For this experiment we opted for columnwise selection of active parameters. This transition from blockwise to columnwise selection was necessary to maintain comparable memory usage across methods, as the number of trainable parameters in LoRA with rank 8 is approximately 2.5 times fewer than the number of parameters in any RoBERTa matrix. For the same reason, we did not include comparisons with BAdam (Luo et al., 2024) in this setup.

The results are presented in Table 6. Since the LoRA setup adds trainable adapters only to the Q and V matrices, while the GaLore code uses all modules as projectable parameters, we conducted experiments in both setups. The results demonstrate that FRUGAL significantly outperforms GaLore and shows comparable results to LoRA.

As in Section 6.1, we conducted additional experiments with FRUGAL using $\rho = 0.0$. In this setup, only the classification head is trained using AdamW, while the embedding parameters remain frozen, and the remaining parameters are trained using signSGD. The results demonstrate that this training approach barely compromises performance compared to FRUGAL with rank 8, and still outperforms GaLore.

Similar to our findings in Section 6.1, we observe that the classification head parameters are particularly sensitive to

the choice of optimizer, which can be seen in Table 19 where the model's performance significantly deteriorates when using signSGD for classification head optimization.

### 7.2. Fine-tuning LLaMA on commonsense reasoning

In addition to our experiments with RoBERTa, we conducted experiments on LLM fine-tuning to evaluate our framework's performance in this practically important area. To assess this capability, we chose LLaMA 3.1-8B (Dubey et al., 2024) and commonsense reasoning benchmark, as this domain represents a fundamental capability requiring both factual knowledge and logical inference. This benchmark includes 8 subtasks each containing its own training and test splits. Following the setup from Hu et al. (2023) we train the model on a single combined Commonsense170K dataset (Hu et al., 2023). We refer readers to the original paper for detailed descriptions of these tasks and the construction of the Commonsense170K dataset.

Following the experimental protocol from Hu et al. (2023), we apply memory-efficient methods to the same parameter subsets: the Q, K, V, Up, and Down projection matrices. We used the same hyperparameter configuration as in the original work, except for the learning rate, which we varied across [5e-6, 1e-5, 2e-5, 5e-5, 1e-4, 2e-4] for each algorithm to ensure optimal performance. The results in Table 7 show that FRUGAL again slightly outperforms both LoRA and GaLore in the average accuracy across all 8 tasks. Remarkably, our method attains this advantage even with $\rho = 0$ (effectively signSGD), requiring zero memory for optimizer state storage while delivering better accuracy.

## 8. Conclusion

In this work, we introduce a new memory-efficient optimization framework, FRUGAL. Within this framework, the optimization space is divided into two subspaces: the first is updated using a state-full algorithm such as Adam, while the second is updated using a state-free algorithm such as signSGD. We prove theoretical convergence guarantees for our framework with SGDM serving as the state-full algorithm and SGD as the state-free algorithm. In experiments involving pre-training and fine-tuning of language models, FRUGAL outperforms other approaches.

## Acknowledgements

The work on the final version was conducted at Moscow Institute of Physics and Technology and was supported by a grant for research center in the field of artificial intelligence, provided by the Ministry of Economic Development of the Russian Federation (agreement No. 139-15-2025-013, dated June 20, 2025, subsidy identifier 000000C313925P4B0002). The work was partially conducted while Philip Zmushko visited Mohamed bin Zayed University of Artificial Intelligence (MBZUAI).

## Impact Statement

Our work makes large-scale model training more accessible to the broader research community by reducing memory requirements. This reduction in computational demands not only decreases the financial cost of training but also potentially lowers the environmental impact. The proposed methods enable researchers with limited resources to participate in large-scale machine learning research, potentially accelerating progress in the field through wider community involvement.

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

# A. Experimental setups

This section describes the main setups used in the experiments and presents additional experiments.

To begin, we introduce the hyperparameter density $\rho$. This hyperparameter represents the fraction of the total space in Linear layers that is updated with a stateful optimizer. For GaLore, this parameter is equal to $\rho = r/h$, where $r$ is the projection rank, and $h$ is the hidden size of the model. For the RandK projection, this parameter can be expressed as $1 - s$, where $s$ means sparsity. For BAdam and FRUGAL with the blockwise update, this parameter denotes the ratio of the number of active blocks $a_{\text{block}}$ to the total number of blocks $p$, that is, $\rho = a_{\text{block}}/p$. When using FRUGAL with the column-wise update, as in Section 7, $\rho$ is equal to the ratio of the number of active columns $a_{\text{column}}$ to their total number $h$, i.e., $\rho = a_{\text{column}}/h$.

## A.1. Pre-training setup

We adopt a LLaMA-based architecture with RMSNorm (Zhang & Sennrich, 2019) and SwiGLU (Shazeer, 2020) activations on the C4 dataset. Following Zhao et al. (2024a), we trained using a batch size of 512 sequences, sequence length of 256, weight decay of 0, and no gradient clipping. We used T5 tokenizer, since it also was trained on C4 with dictionary size equal to 32k. The update frequency $T$ is set to 200.

Since, unlike GaLore, we consider not only matrix projections, we decided to generalize the concept of rank $r$. Instead, we use density $\rho$, which represents the proportion of Linear layer parameters in the state-full subspace. Thus, for SVD-like projection as in GaLore, the density equals $\rho = r/h$, where $h$ denotes the hidden dimension of the model. We also should point out that similarly to Zhao et al. (2024a), we keep Embeddings, RMSNorms, and Output layer in the state-full subspace throughout the training and don't reset the optimizer state for them.

We used standard Adam hyperparameters: $\beta_1 = 0.9, \beta_2 = 0.999, \varepsilon = 1e - 8$. For all methods except GaLore, we selected the learning rate equal to the optimal learning rate for Adam, which we determined through a grid search among values $[1e - 4, 3e - 4, 1e - 3, 3e - 3]$. FRUGAL's learning rate for the state-free optimizer was set equal to that for the state-full optimizer for simplicity and ease of tuning. For a fair comparison with GaLore (Zhao et al., 2024a), we conducted experiments with two learning rate values: 1) the one specified by the authors in the original paper and 2) the optimal learning rate for Adam, as used for other methods. We did this because the learning rate in the original paper could have been optimized for a different number of iterations.

To match the learning rate changes in the first steps of our training with Zhao et al. (2024a), we used a cosine learning rate schedule with restarts, with a warmup of 10% of the steps in a cycle length, and decay of the final learning rate down to 10% of the peak learning rate. To verify that our results are not sensitive to the choice of scheduler, we repeated the experiments for LLaMA-130M with other schedulers. The results for constant with warm-up and cosine (one cycle) with warm-up schedulers can be found in Tables 15 and 16.

For pre-training GPT-2 124M (Radford et al., 2019) we followed the setup described above except for the tokenizer. We utilized the GPT-2 original tokenizer, with 50257 vocabulary size. The results are presented in Table 12.

*Table 8.* Comparison of validation perplexity and memory estimation for various optimization methods across LLaMA model scales trained on C4 **with $\beta_2 = 0.95$**. We also indicate the additional memory overhead introduced by the optimization algorithm. The values are calculated assuming that each float value occupies 4 bytes (float32). $\rho$ denotes the proportion of the Linear layer parameters in the state-full subspace. Note that Embeddings, RMSNorms, and Output layer are always trained with AdamW.

|  | 60M | 130M | 350M |
|---|---|---|---|
| AdamW | 23.51 (0.43G) | 18.32 (1.00G) | 14.57 (2.74G) |
| GaLore, $\rho = 0.25$ | 26.66 (0.30G) | 21.03 (0.54G) | 16.79 (1.10G) |
| BAdam, $\rho = 0.25$ | 25.40 (0.29G) | 20.17 (0.52G) | 16.54 (1.05G) |
| FRUGAL, $\rho = 0.25$ | **24.07** (0.29G) | **18.79** (0.52G) | **14.96** (1.05G) |
| FRUGAL, $\rho = 0.0$ | 24.54 (0.24G) | 19.11 (0.37G) | 15.20 (0.49G) |
| Training tokens | 20B | 20B | 24B |
| Number of iterations | 200k | 200k | 240k |

*Table 9.* Perplexity of LLaMA-130M models pre-trained on C4 using pure bfloat16 format both for model weights and optimizer statistics.

| Method | 100k iterations |
|---|---|
| Adam | 21.88 |
| GaLore, $\rho = 0.25$ | 24.19 |
| BAdam, $\rho = 0.25$ | 25.03 |
| FRUGAL, $\rho = 0.25$ | 23.17 |
| FRUGAL, $\rho = 0.0$ | **22.64** |

*Table 10.* Perplexity of LLaMA-130M models pre-trained on C4 for 200k steps with different state-free optimizers for FRUGAL.

| Method | State-free optimizer | Validation perplexity |
|---|---|---|
| Adam | — | 18.13 |
| FRUGAL, $\rho = 0.25$ | signSGD | **18.60** |
| FRUGAL, $\rho = 0.25$ | SGD | 19.11 |

*Table 11.* Perplexity of LLaMA-130M models pre-trained on C4 with Lion as state-full optimizer for 200k steps.

| Method | 200k |
|---|---|
| Adam | 18.13 |
| Lion | 18.55 |
| GaLore (+ Lion), $\rho = 0.25$ | 21.65 |
| FRUGAL (+ Lion), $\rho = 0.25$ | **18.89** |

*Table 12.* Validation perplexity of GPT-2 124M model pre-trained on C4 for 200k steps with various optimization methods.

| Method | Validation perplexity |
|---|---|
| Adam | 21.94 |
| GaLore, $\rho = 0.25$ | 25.84 |
| BAdam, $\rho = 0.25$ | 25.43 |
| FRUGAL, $\rho = 0.25$ | **23.23** |
| FRUGAL, $\rho = 0.0$ | 25.04 |

*Table 13.* Perplexity of LLaMA-130M models pre-trained on C4 for 200k iterations using FRUGAL with $\rho = 1/3$ and different Block update strategy, taken from Luo et al. (2024).

| Method | Perplexity |
|---|---|
| Random | **18.50** |
| Ascending | 18.54 |
| Descending | **18.50** |

*Table 14.* Perplexity of LLaMA-130M models pre-trained on C4 for 200k iterations (20B tokens) using FRUGAL with $\rho = 0.25$ and different update frequency $T$.

| Update frequency $T$ | Perplexity |
|---|---|
| 10 | 18.82 |
| 20 | 18.73 |
| 50 | 18.69 |
| 100 | 18.65 |
| 200 | **18.60** |
| 500 | **18.60** |
| 1000 | 18.61 |

*Table 15.* Perplexity of LLaMA-130M models pre-trained on C4 using constant scheduler with warm-up at various training iterations.

| Method | 100k | 200k |
|---|---|---|
| Adam | 19.51 | 18.51 |
| GaLore, $\rho = 0.25$ | 22.63 | 21.03 |
| BAdam, $\rho = 0.25$ | 22.31 | 20.66 |
| FRUGAL, $\rho = 0.25$ | **19.97** | **18.85** |
| FRUGAL, $\rho = 0.0$ | 20.33 | 19.14 |

*Table 16.* Perplexity of LLaMA-130M models pre-trained on C4 using cosine scheduler with warm-up at various training iterations.

| Method | 100k | 200k |
|---|---|---|
| Adam | 19.38 | 17.95 |
| GaLore, $\rho = 0.25$ | 22.30 | 20.60 |
| BAdam, $\rho = 0.25$ | 22.35 | 20.07 |
| FRUGAL, $\rho = 0.25$ | **19.62** | **18.16** |
| FRUGAL, $\rho = 0.0$ | 19.83 | 18.34 |

### A.2. RoBERTa fine-tuning setup

The batch size and learning rate values used for FRUGAL in the experiments from Table 6 are presented in Table 18. In all experiments, we set the learning rate for the state-free optimizer to $1/10$ of the learning rate of the state-full optimizer. Other hyperparameters, such as scheduler, number of epochs, maximum sequence length, and warmup ratio, were taken from Hu et al. (2021).

*Table 17.* Perplexity of LLaMA-130M models pre-trained on C4 for 200k iterations (20B tokens) using `FRUGAL` with different density $\rho$.

| | FRUGAL | | | | | | | |
|---|---|---|---|---|---|---|---|---|
| $\rho$ | 1.0 (**Adam**) | 0.5 | 0.33 | 0.25 | 0.125 | 0.0625 | 0.0 | **signSgd** |
| Perplexity | 18.13 | 18.40 | 18.50 | 18.63 | 18.71 | 18.80 | 18.90 | 33.22 |

*Table 18.* Hyperparameters of fine-tuning RoBERTa-base for `FRUGAL`.

| | MNLI | SST-2 | MRPC | CoLA | QNLI | QQP | RTE | STS-B |
|---|---|---|---|---|---|---|---|---|
| Batch Size | 128 | 128 | 16 | 256 | 256 | 128 | 32 | 16 |
| State-full Learning Rate | 5E-05 | 5E-05 | 2E-04 | 5E-04 | 1E-04 | 5E-05 | 2E-04 | 1E-04 |
| State-free lr multiplier | | | | 0.1 | | | | |
| Rank/Density | | | $r = 8$ / | $r = 0$ $(\rho = 0)$ | | | | |

We also present a comparison between fine-tuning using `FRUGAL` with $\rho = 0.0$ and full fine-tuning using signSGD. Essentially, the only difference is that in the second case, the classification head is updated with signSGD instead of Adam. The results in Table 19 show that the classification head is extremely sensitive to the optimizer type, and switching the optimizer significantly drops the accuracy.

*Table 19.* Results of fine-tuning RoBERTa-Base on several tasks from GLUE. The left column indicates which modules were trained using the state-full optimizer Adam. The remaining modules, except for the frozen Embedding layer, were trained using the state-free signSGD.

| Method | | SST2 | QNLI | QQP |
|---|---|---|---|---|
| **Classification head** (corresponds to the `FRUGAL` with $\rho = 0.0$) | | **94.9**$_{\pm.2}$ | 92.8$_{\pm.1}$ | 91.3$_{\pm.1}$ |
| **None** (corresponds to the fine-tuning using signSGD) | | 89.7 | 81.6 | 74.3 |

## B. Comparison with Concurrent Methods

In this section, we present a comparison with several concurrent works — namely, Fira (Chen et al., 2024a), LDAdam (Robert et al., 2024), and Adamem (Vyas et al., 2024) — which we discovered only during the final stages of our project. Due to this timing, we did not have the opportunity to conduct comprehensive experimental comparisons with them as we did with other baselines in Table 2.

### B.1. Algorithmic comparison

We begin by comparing these methods with `FRUGAL` from an algorithmic perspective.

**Adamem.** Similarly to our framework, Vyas et al. (2024) divides the gradient into two parts: the first part is a projection of the gradient onto the top SVD subspace, while the second part is the residual outside this subspace. The first part is used to update momentum, which is then fed into Adafactor's preconditioner Shazeer & Stern (2018), while the second part is fed directly into a one-sided Adafactor preconditioner Vyas et al. (2024). The outputs of these preconditioners are then used for the final parameter update. Although the core idea of splitting the gradient into two orthogonal subspaces is analogous to Algorithm 1, Adamem implements *only one possible variant* for choosing subspace and for updates of each component. Thus, Adamem represents a *special case of* `FRUGAL`: with SVD-based projection, Adafactor with momentum as the state-full optimizer, and one-sided Adafactor as the state-free optimizer.

**Fira.** Chen et al. (2024a) also apply the idea of decomposing the full-rank gradient into a low-rank subspace gradient and a residual gradient. Similarly to GaLore, the low-rank part is obtained through SVD and passed through an AdamW step; however, unlike GaLore, the residual part is not discarded, but is also used in the update in an SGD-like format without using additional optimizer state. An important feature that Fira introduces is norm-based scaling: this technique adaptively scales the per-column learning rate for the residual part by $\frac{\|\psi_t(R_t)\|}{\|R_t\|}$ where $\psi_t$ denotes the AdamW update rule. In addition, for

*Table 20.* Comparison of validation perplexity for AdamW, FRUGAL and AdaMeM across LLaMA model scales trained on C4.

|  | 60M | 130M | 350M |
|---|---|---|---|
| AdamW | 22.73 | 18.13 | 14.43 |
| AdaMeM, $\rho = 0.25$ | 23.81 | 18.99 | 15.10 |
| FRUGAL, $\rho = 0.25$ | **23.59** | **18.60** | **14.79** |
| FRUGAL, $\rho = 0.0$ | 24.06 | 18.90 | 15.03 |
| Training tokens | 20B | 20B | 24B |
| Number of iterations | 200k | 200k | 240k |

enhanced training stability, Fira proposes replacing standard gradient clipping with a norm-growth limiter, which transforms abrupt gradient spikes into gradual, smooth increases.

Thus, the key difference between Fira and FRUGAL is that the state-free update sophisticatedly utilizes information from the state-full update. However, Fira has several limitations. First, Fira relies on computing SVD for projection onto the low-rank subspace, which introduces additional memory and computational overhead, with the computational burden becoming more pronounced as model size increases (see Section 4 and Appendix C. Moreover, it is not readily obvious how to propose an alternative to SVD to eliminate this overhead, because, for example, block projection from BAdam (Luo et al., 2024) cannot work with norm-based scaling (Chen et al., 2024a) by construction. Second, following GaLore, Fira continues using the old optimizer state after updating the projection, which should be suboptimal (see Appendix D). Furthermore, according to the original paper Chen et al. (2024a), the reason for instability in standard gradient clipping may be precisely the moments of low-rank subspace transitions.

We also would like to note separately that neither Fira (Chen et al., 2024a) nor Adamem (Vyas et al., 2024) provide theoretical convergence proofs. In contrast, *we provide a proof that recovers the best known convergence rate under conventional assumptions.*

**LDAdam.** Notably, unlike Fira and Adamem, Robert et al. (2024) identify the addition of momentum and gradient from different subspaces as mathematically incorrect (similarly to Appendix D). To resolve this inconsistency, Robert et al. (2024), propose to reproject the previous optimizer state (similarly to Algorithm 1). Interestingly, LDAdam differs from FRUGAL, Adamem, and Fira in that each individual step remains low-rank, but unlike GaLore and BAdam, the remaining information is *not discarded but preserved in an error feedback buffer*, thus allowing LDAdam to have convergence guarantees and demonstrate high performance.

However, LDAdam also has a drawback. The method requires updating the projection at every step; and although Robert et al. (2024) proposed replacing the expensive SVD decomposition with significantly cheaper block power iteration, the necessity of performing this operation at every step causes a slowdown of $\geq 15\%$.

### B.2. Experimental comparison

We also present a preliminary experimental comparison of FRUGAL with Adamem (Vyas et al., 2024), Fira (Chen et al., 2024a), and LDAdam (Robert et al., 2024).

**Adamem.** We compared FRUGAL with our reimplementation of Adamem (Vyas et al., 2024) on pre-training LLaMA model of 3 sizes — 60M, 130M, and 350M — in the same setup as in Section 6.1 and Table 2. Unique Adamem hyperparameters were taken from Vyas et al. (2024). The results presented in Table 20 show that while Adamem performs significantly better than GaLore due to utilizing information from the gradient residual, it still falls slightly short of FRUGAL. However, at this point, we are not ready to claim whether this is a consequence of a less favorable choice of state-full and state-free update rules or potentially suboptimal hyperparameter settings for our experimental setup.

**Fira and LDAdam** For experiments with Fira (Chen et al., 2024a) and LDAdam (Robert et al., 2024), we deviated from the setup in Appendix A and Table 2. This modification was necessary since our main setup following GaLore (Zhao et al., 2024a) does not use gradient clipping, while its counterpart—the norm-growth limiter—is critically important for Fira's performance. Additionally, we observed that Fira significantly benefits from using weight decay, so for the experiments

*Table 21.* Comparison of validation perplexity for AdamW, FRUGAL Fira and LDAdam across LLaMA model scales trained on C4. We also report approximate slowdown comparing to AdamW (taken from original papers).

|  | **Approximate Slowdown** | 130M | 350M |
|---|---|---|---|
| AdamW | - | 17.52 | 13.81 |
| Fira, $\rho = 0.25$ | 10% | 17.58 | 13.74 |
| LDAdam, $\rho = 0.25$ | 15% | **17.54** | **13.54** |
| FRUGAL, $\rho = 0.25$ | **0%** | 17.58 | 13.99 |

in Table 21 we enabled both gradient clipping (norm-growth limiter for Fira) and weight decay. The results show that all three methods achieve performance very close to AdamW. Surprisingly, LDAdam even outperforms the full-rank AdamW baseline for the 350M model (which, however, may be a result of insufficient hyperparameter grid search). Fira performs slightly better than FRUGAL.

However, we would like to note that these improvements are valid in terms of the *number of iterations*, and not in terms of *wall clock time*, where both Fira and LDAdam introduce *noticeable time overhead*. In the same table, we report the approximate wall-clock slowdown of the methods compared to plain AdamW. We provide approximate values taken from the original papers because the actual slowdown can vary significantly depending on the setup (number of workers in training, GPU type, etc.). One can observe that the slowdown due to the need to perform additional heavyweight operations during the optimizer step can negate the advantages gained from a more carefully designed update rule. Thus, practitioners should choose the algorithm for their specific use case.

## C. Memory estimation

In this section, we will examine memory requirements for different projection types using the LLaMA-like architecture as an example and show that RandK, column-wise, and blockwise projections result in approximately the same amount of additional memory for a given density value $\rho$ Appendix A. In contrast, the semi-orthogonal projection matrix (GaLore-like) requires a slightly larger value in this setup. Recall that we follow the setup from Zhao et al. (2024a), where Embeddings, RMSNorms, and Output layer remain in the state-full subspace throughout the training, so the projection does not interact with them, and they give the same memory overhead for all projection methods.

Let the number of parameters in the remaining projectable parameters be $P$. Then, training using Adam gives an additional overhead of $2P$ float values for storing $m$ and $v$ for each parameter. Now, let's consider blockwise and column-wise projections and suppose we want to achieve a density $\rho$. For blockwise, we take round($\rho \cdot L$) layers, where $L$ is the total number of transformer layers, and for column-wise, we take round($\rho \cdot k$) columns for each matrix of size $n \times k$. Since the memory required to store block or column indices is negligible compared to other costs, we find that the total size of the optimizer state when using Adam as a state-full optimizer will be $2\rho \cdot P$, with an adjustment for rounding.

In the case of RandK projection, we have the same $2\rho \cdot P$ float values $M$ and $V$ in the optimizer state. However, we must also know the current indices corresponding to these values. On the other hand, it is widely known that if one needs to save a set of random values, they don't need to store all these values - it's sufficient to store only the seed from which they were generated. Thus, for RandK, the total memory also equals $2\rho \cdot P$.

If we recalculate this considering a specific LLaMA-like architecture, each layer consists of 7 matrices: 4 matrices of size $h \times h$ (Query, Key, Value, Output) and 3 matrices of size $h \times h_{ff}$ (Gate, Down, Up), where $h$ is the hidden size of the model, and $h_{ff}$ is the FFN hidden size. In the LLaMA architecture, it's typically:

$$h_{ff} = 4h \cdot \frac{2}{3} = \frac{8}{3}h.$$

Then, the amount of memory for RandK projection (and consequently for all others mentioned above) is:

$$2 \cdot (4 \cdot (\rho h^2) + 3 \cdot (\rho \cdot h \cdot h_{ff})) = 2 \cdot (4 \cdot \rho h^2 + 3 \cdot (\frac{8}{3}\rho \cdot h^2)) = 24\rho \cdot h^2$$

for each layer on average (2 corresponds to the number of matrices $M$ and $V$).

In the case of a GaLore-like semi-orthogonal projection matrix, the situation is as follows. We have projections onto a low-rank subspace of rank $r$, where $r = \text{round}(\rho \cdot h)$. Then, for Query, Key, Value, and Output projections, we need to store $\boldsymbol{P}, \boldsymbol{M}, \boldsymbol{V} \in \mathbb{R}^{h \times r}$, and for Gate, Down and Up projections either $\boldsymbol{P} \in \mathbb{R}^{h \times r}, \boldsymbol{M}, \boldsymbol{V} \in \mathbb{R}^{h_{ff} \times r}$, or $\boldsymbol{P} \in \mathbb{R}^{h_{ff} \times r}, \boldsymbol{M}, \boldsymbol{V} \in \mathbb{R}^{h \times r}$. Since the second option requires less memory, it is used by default in (Zhao et al., 2024a) and, therefore, in FRUGAL, too. Then, the total memory requirements are:

$$4 \cdot (3 \cdot rh) + 3 \cdot (2 \cdot r \cdot h + r \cdot h_{ff}) = 12rh + 6rh + 3rh_{ff} = (12 + 6 + 3 \cdot \frac{8}{3})rh = 26\rho h^2.$$

To sum up, RandK, column-wise and blockwise projection requires $2\rho P$ additional memory, while semi-orthogonal projection (GaLore-like) requires $\frac{26}{24} \cdot 2\rho P = \frac{13}{12} \cdot 2\rho P$ additional memory.

Let's recall that in addition to this, SVD requires additional computation, which can take up to 10% as the model size increases (Zhao et al., 2024a). Therefore, for our method, we settled on blockwise projection.

## D. Optimizer state management

In this section, we would like to propose some modifications to the GaLore algorithm. These modifications are also used in our framework as SVD projection.

Specifically, we want to consider the projection of the state when changing the active subspace. In GaLore (Zhao et al., 2024a), when updating the projection, the optimizer states $M$ and $V$ do not change. This results in new projected gradients and old $M$ and $V$ being in different subspaces. This implementation has little effect on the result with large values of update frequency $T$, as the values of $M$ and $V$ from the previous subspace decay exponentially quickly. However, more frequent changes $T$ significantly affect the result. We hypothesize that this is why in Zhao et al. (2024a) the model quality degraded so significantly when $T$ was decreased, while as seen in Table 14, FRUGAL experiences much less degradation.

There are two different ways to overcome this obstacle: either project the state back to full-rank space or reset the state before a new round. However, the first option may be challenging in the case of arbitrary projection. Specifically, while it's possible to project momentum back to full-rank space (see Alg. 2 in Hao et al. (2024)), the same cannot be easily done with variance because its values depend quadratically on the projection matrix. However, the projection of variance will also be trivial if the set of basis vectors for the projection is fixed, which is true, for example, for coordinate projection with RandK.

To demonstrate the effectiveness of this improvement, we provide a toy example. We consider a quadratic minimization problem of $\|W\|^2, W \in \mathbb{R}^{10 \times 10}$. For optimization, we use GaLore-like SGDM and GaLore-like SGDM with Momentum state projection. This projection is similar to Alg. 2 from (Hao et al., 2024), except we additionally normalize the new momentum by the ratio of norms before and after re-projection to preserve momentum mass. We use ranks of 3 and 6, and an update frequency $T = 10$ and plot mean and standard deviation across 5 independent runs. The results are presented in Figure 3. As can be seen, the variant with state projection converges much faster.

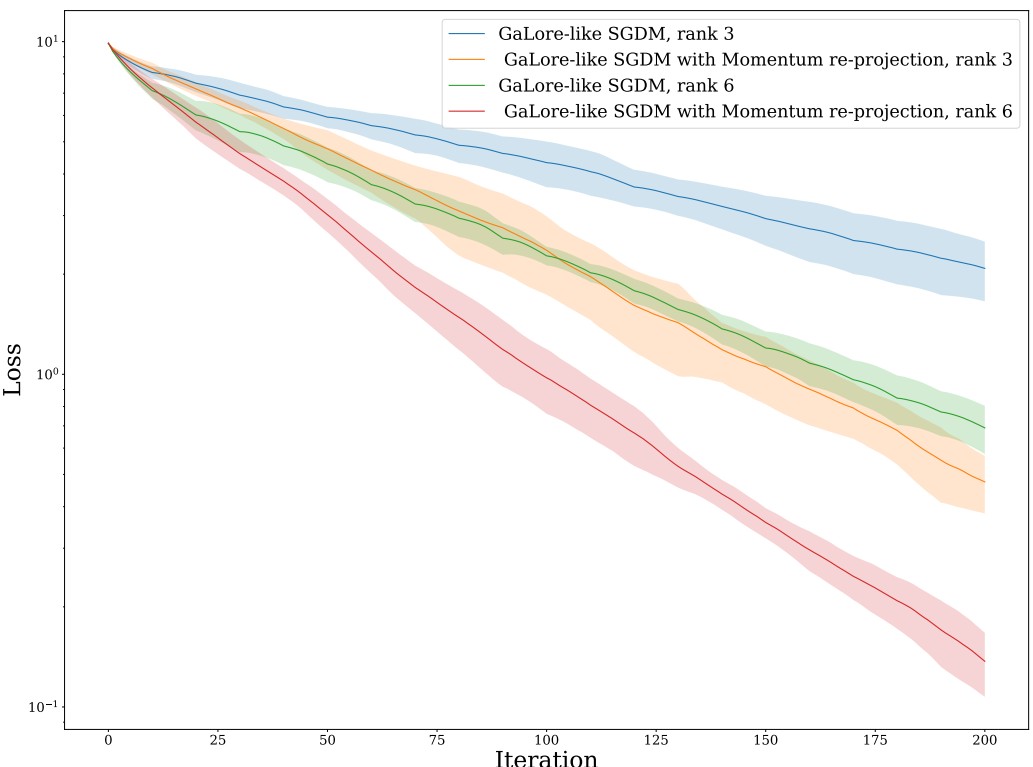

*Figure 3.* Toy example of solving quadratic minimization problem with GaLore-like SGDM with and without re-projection of optimizer state. Algorithm with re-projection converges much faster.

# E. Convergence Theory

Firstly, we provide ommited definition of $L$-smooth function.

**Definition E.1.** We say that $f : \mathbb{R}^d \to \mathbb{R}$ is $L-$smooth with $L \geq 0$, if it is differentiable and satisfies

$$f(y) \leq f(x) + \langle \nabla f(x), y - x \rangle + \frac{L}{2}\|y - x\|^2, \forall x, y \in \mathbb{R}^d.$$

Below, we provide an equivalent formulation of Algorithm 2 that enables us to use the proof of the similar structure to SGDM momentum analyis of (Liu et al., 2020).

---

**Algorithm 3** FRUGAL(SGDM, SGD): Equivalent to Algorithm 2 for constant step size

---

**Input:** momentum weight $\beta \in [0, 1)$, initialization $x^1 \in \mathbb{R}^d$ and $m^0 = 0$, step sizes $\{\alpha_k := \alpha > 0\}_{k=1}^K$, momentum set $J_k \subset [d]$ for $k = 1, 2 \dots$

1: **for** $k = 1, 2, \dots$ **do**

2:     Compute stochastic gradient $\tilde{g}^k \leftarrow \nabla f_{\zeta^k}(x^k)$

3:     Update momentum vector $\tilde{m}_j^k \leftarrow (1 - \beta)\tilde{g}_j^k + \beta \begin{cases} \tilde{m}_j^{k-1} & \text{if } j \in J_k, \\ 0 & \text{otherwise} \end{cases}$

4:     Update iterate $x^{k+1/2} \leftarrow x^k - \alpha\tilde{m}^k$

5:     $x_j^{k+1} \leftarrow \begin{cases} \frac{x_j^{k+1/2}}{1-\beta} - \frac{\beta x_j^k}{1-\beta} & \text{if } j \notin J_{k+1}, \\ x_j^{k+1/2} & \text{otherwise} \end{cases}$

6: **end for**

---

Next, we present several key ingredients of the proof. Firstly, we can express the momentum term $\tilde{m}_j^k$ as

$$\tilde{m}_j^k = (1 - \beta) \sum_{i=t_j^k}^k \beta^{k-i}\tilde{g}_j^i, \tag{3}$$

where $t_j^k := \max_{t \leq k}\{j \notin J_t\}$, i.e., the last time when the momentum buffer was released. We denote

$$m_j^k = (1 - \beta) \sum_{i=t_j^k}^k \beta^{k-i}g_j^i, \tag{4}$$

Using this notation, we proceed with two lemmas, one showing variance reduction effect of momentum, the other boundess of momentum bias.

**Lemma E.2.** *Under Assumption 5.1, the update vector $\tilde{m}^k$ in Algorithm 3 satisfies*

$$\mathbb{E}\left[\left\|\tilde{m}^k - m^k\right\|^2\right] \leq \frac{1 - \beta}{1 + \beta}\sigma^2.$$

*Proof.* Since $\tilde{m}_j^k = (1 - \beta)\sum_{i=t_j^k}^k \beta^{k-i}\tilde{g}_j^i$, we have

$$\mathbb{E}\left[\left\|\tilde{m}^k - m^k\right\|^2\right] = \sum_{j\in[d]} \mathbb{E}\left[\left\|\tilde{m}_j^k - m_j^k\right\|^2\right]$$

$$\leq (1 - \beta)^2 \sum_{j\in[d]} \mathbb{E}\left[\left\|\sum_{i=t_j^k}^k \beta^{k-i}(\tilde{g}_j^i - g_j^i)\right\|^2\right].$$

Moreover, since $\zeta^1, \zeta^2, ..., \zeta^k$ are independent random variables (item 3 of Assumption 5.1), we can use conditional expectation to show that $\mathbb{E}\left[(\tilde{g}_j^{i_1} - g_j^{i_1})(\tilde{g}_j^{i_2} - g_j^{i_2})\right] = 0$ for $i_1 \neq i_2$. Therefore,

$$\mathbb{E}\left[\left\|\tilde{m}^k - m^k\right\|^2\right] \leq (1-\beta)^2 \sum_{j\in[d]} \mathbb{E}\left[\sum_{i=t_j^k}^{k} \beta^{2(k-i)}\|\tilde{g}_j^i - g_j^i\|^2\right]$$

$$\leq \frac{1-\beta}{1+\beta} \sum_{j\in[d]} \mathbb{E}\left[(1-\beta^{2(k-t_j^k+1)})\right]\sigma_j^2$$

$$\leq \frac{1-\beta}{1+\beta} \sum_{j\in[d]} \sigma_j^2 = \frac{1-\beta}{1+\beta}\sigma^2.$$

$\square$

**Lemma E.3.** *Under Assumption 5.1, the update vector $\tilde{m}^k$ in Algorithm 3 further satisfies*

$$\mathbb{E}\left[\sum_{j\in J_k}(1-\beta^{k_j})^2\left\|\frac{m_j^k}{(1-\beta^{k_j})} - g_j^k\right\|^2\right] \leq p_{\max}^k \mathbb{E}\left[\sum_{i=1}^{k-1} a_{k,i}\|x^{i+1} - x^i\|^2\right],$$

*where $k_j = k - t_j^k + 1$, and*

$$a_{k,i} = L^2\beta^{k-i}\left(k - i + \frac{\beta}{1-\beta}\right). \tag{5}$$

*Proof.* Let $\Pr_{k-1}[j \in J_k] = p_j^k$ and $p_{\max}^k := \max_{j\in[d]}\{p_j^k\}$. Then,

$$\mathbb{E}\left[\sum_{j\in J_k}(1-\beta^{k_j})^2\left\|\frac{m_j^k}{(1-\beta^{k_j})} - g_j^k\right\|^2\right] = \mathbb{E}\left[\sum_{j\in J_k}(1-\beta^{k_j})^2\left\|\frac{1-\beta}{1-\beta^{k_j}}\sum_{i=t_j^k}^{k}\beta^{k-i}(g_j^i - g_j^k)\right\|^2\right]$$

$$= (1-\beta)^2\mathbb{E}\left[\sum_{j\in J_k}\sum_{i,l=t_j^k}^{k}\langle\beta^{k-i}(g_j^k - g_j^i), \beta^{k-l}(g_j^k - g_j^l)\rangle\right]$$

$$\leq (1-\beta)^2\mathbb{E}\left[\sum_{j\in J_k}\sum_{i,l=1}^{k}\beta^{2k-i-l}\left(\frac{1}{2}\|g_j^k - g_j^i\|^2] + \frac{1}{2}\|g_j^k - g_j^l\|^2\right)\right]$$

$$= (1-\beta)^2\mathbb{E}\left[\sum_{j\in J_k}\sum_{i=1}^{k}\left(\sum_{l=1}^{k}\beta^{2k-i-l}\right)\frac{1}{2}\mathbb{E}[\|g_j^k - g_j^l\|^2]\right]$$

$$+ (1-\beta)^2\mathbb{E}\left[\sum_{j\in J_k}\sum_{l=1}^{k}\left(\sum_{i=1}^{k}\beta^{2k-i-l}\right)\frac{1}{2}[\|g_j^k - g_j^i\|^2]\right]$$

$$= (1-\beta)^2\mathbb{E}\left[\sum_{j\in J_k}\sum_{i=1}^{k}\frac{\beta^{k-i}(1-\beta^{k_j})}{1-\beta}\|g_j^k - g_j^i\|^2\right]$$

$$\leq (1-\beta)\mathbb{E}\left[\sum_{j\in J_k}\sum_{i=1}^{k}\beta^{k-i}\|g_j^k - g_j^i\|^2\right],$$

$$\leq (1-\beta)p_{\max}^k\mathbb{E}\left[\sum_{i=1}^{k}\beta^{k-i}\|g^k - g^i\|^2\right],$$

where we applied Cauchy-Schwarz to the first inequality.

By applying triangle inequality and the smoothness of $f$ (item 1 in Assumption 5.1), we further have

$$\mathbb{E}\left[\sum_{j \in J_k}(1-\beta^{k_j})^2\left\|\frac{m_j^k}{(1-\beta^{k_j})}-g_j^k\right\|^2\right] \leq (1-\beta)p_{\max}^k\mathbb{E}\left[\sum_{i=1}^k \beta^{k-i}(k-i)\sum_{l=i}^{k-1}\|g^{l+1}-g^l\|^2\right]$$

$$\leq \mathbb{E}\left[\sum_{l=1}^{k-1}\left((1-\beta)p_{\max}^k L^2 \sum_{i=1}^l \beta^{k-i}(k-i)\right)\|x^{l+1}-x^l\|^2\right].$$

Therefore, by defining $a'_{k,l}=(1-\beta)L^2\sum_{i=1}^l \beta^{k-i}(k-i)$, we get

$$\mathbb{E}\left[\sum_{j \in J_k}(1-\beta^{k_j})^2\left\|\frac{m_j^k}{(1-\beta^{k_j})}-g_j^k\right\|^2\right] \leq p_{\max}^k\mathbb{E}\left[\sum_{l=1}^{k-1}a'_{k,l}\|x^{l+1}-x^l\|^2\right]. \tag{6}$$

Furthermore, $a'_{k,j}$ can be calculated as

$$a'_{k,l}=L^2\beta^k\left(-(k-1)-\frac{1}{1-\beta}\right)+L^2\beta^{k-l}\left(k-l+\frac{\beta}{1-\beta}\right). \tag{7}$$

Notice that

$$a'_{k,l}<a_{k,l}:=L^2\beta^{k-l}\left(k-l+\frac{\beta}{1-\beta}\right). \tag{8}$$

Combining this with equation 6, we arrive at

$$\mathbb{E}\left[\sum_{j \in J_k}(1-\beta^{k_j})^2\left\|\frac{m_j^k}{(1-\beta^{k_j})}-g_j^k\right\|^2\right] \leq p_{\max}^k\mathbb{E}\left[\sum_{i=1}^{k-1}a_{k,i}\|x^{i+1}-x^i\|^2\right],$$

where

$$a_{k,i}=L^2\beta^{k-i}\left(k-i+\frac{\beta}{1-\beta}\right).$$

$\square$

From Lemma E.3, we know that the distance of the non-stochastic momentum from $g^k$ is bounded by the weighted sum of past successive iterate differences. Furthermore, the coefficients $a_{k,i}$ decays exponentially in $\beta$.

Therefore, we use the following Lyapunov function

$$L^k=\left(f(z^k)-f^\star\right)+\sum_{i=1}^{k-1}c_i\|x^{k+1-i}-x^{k-i}\|^2. \tag{9}$$

for some positive $c_i$ that we specify later. As it is common for convergence theory of SGDM to analyze an auxiliary sequence $z^k$ defined as

$$z_j^k=\begin{cases}x_j^k & k=1,\\ \frac{1}{1-\beta}x_j^{k-1/2}-\frac{\beta}{1-\beta}x_j^{k-1} & k\geq 2,\end{cases} \tag{10}$$

which behaves more like an SGD iterate, although the stochastic gradient $\tilde{g}^k$ is not taken at $z^k$.

**Lemma E.4.** *Let $x^k$'s be iterates of Algorithm 3, then $z^k$ defined in equation 10 satisfies*

$$z^{k+1}-z^k=-\alpha\tilde{g}^k.$$

*Proof.* We have to consider two different cases. Firstly, if $k = 1$ or $j \notin J_k$, then

$$z_j^{k+1} - z_j^k = \frac{x_j^{k+1/2}}{1-\beta} - \frac{\beta x_j^k}{1-\beta} - x_j^k = \frac{x_j^k - \alpha \tilde{m}_j^k - \beta x_j^k - (1-\beta)x_j^k}{1-\beta} = -\frac{\alpha(1-\beta)\tilde{g}_j^k}{1-\beta} = -\alpha \tilde{g}_j^k.$$

Secondly, if $k \geq 2$, $j \in J_k$, then

$$\begin{aligned}
z_j^{k+1} - z_j^k &= \frac{1}{1-\beta}(x_j^{k+1/2} - x_j^{k-1/2}) - \frac{\beta}{1-\beta}(x_j^k - x_j^{k-1}) \\
&= \frac{1}{1-\beta}(x_j^{k+1/2} - x_j^k) - \frac{\beta}{1-\beta}(x_j^k - x_j^{k-1}) \\
&= \frac{1}{1-\beta}(-\alpha \tilde{m}_j^k) - \frac{\beta}{1-\beta}(-\alpha \tilde{m}_j^{k-1}) \\
&= \frac{1}{1-\beta}(-\alpha \tilde{m}_j^k + \alpha \beta \tilde{m}_j^{k-1}) = -\alpha \tilde{g}_j^k.
\end{aligned}$$

$\square$

Before procceding with the main convergence theory, we require one more proposition that shows descent in objective value.

**Proposition E.5.** *Take Assumption 5.1. Then, for $z^k$ defined in equation 10, we have*

$$\begin{aligned}
\mathbb{E}[f(z^{k+1})] \leq{} & \mathbb{E}[f(z^k)] + \left(-\alpha + \frac{1+\beta^2}{1-\beta}L\alpha^2 + \frac{1}{2}L\alpha^2\right)\mathbb{E}[\|g^k\|^2] \\
& + \left(\frac{\beta^2}{2(1+\beta)} + \frac{1}{2}\right)L\alpha^2\sigma^2 + \frac{L\alpha^2}{1-\beta}\mathbb{E}\left[\sum_{j\in J_k}(1-\beta^{k_j})^2\left\|\frac{m_j^k}{(1-\beta^{k_j})} - g_j^k\right\|^2\right].
\end{aligned} \tag{11}$$

*Proof.* The smoothness of $f$ yields

$$\begin{aligned}
\mathbb{E}_{\zeta^k}[f(z^{k+1})] &\leq f(z^k) + \mathbb{E}_{\zeta^k}[\langle \nabla f(z^k), z^{k+1} - z^k\rangle] + \frac{L}{2}\mathbb{E}_{\zeta^k}[\|z^{k+1} - z^k\|^2] \\
&= f(z^k) + \mathbb{E}_{\zeta^k}[\langle \nabla f(z^k), -\alpha \tilde{g}^k\rangle] + \frac{L\alpha^2}{2}\mathbb{E}_{\zeta^k}[\|\tilde{g}^k\|^2],
\end{aligned} \tag{12}$$

where we have applied Lemma E.4 in the second step.

For the inner product term, we can take full expectation $\mathbb{E} = \mathbb{E}_{\zeta^1}...\mathbb{E}_{\zeta^k}$ to get

$$\mathbb{E}[\langle \nabla f(z^k), -\alpha \tilde{g}^k\rangle] = \mathbb{E}[\langle \nabla f(z^k), -\alpha g^k\rangle],$$

which follows from the fact that $z^k$ is determined by the previous $k-1$ random samples $\zeta^1, \zeta^2, ...\zeta^{k-1}$, which is independent of $\zeta^k$, and $\mathbb{E}_{\zeta^k}[\tilde{g}^k] = g^k$.

So, we can bound

$$\begin{aligned}
\mathbb{E}[\langle \nabla f(z^k), -\alpha \tilde{g}^k\rangle] &= \mathbb{E}[\langle \nabla f(z^k) - g^k, -\alpha g^k\rangle] - \alpha \mathbb{E}[\|g^k\|^2] \\
&\leq \alpha \frac{\rho_0}{2}L^2\mathbb{E}[\|z^k - x^k\|^2] + \alpha \frac{1}{2\rho_0}\mathbb{E}[\|g^k\|^2] - \alpha \mathbb{E}[\|g^k\|^2],
\end{aligned}$$

where $\rho_0 > 0$ can be any positive constant (to be determined later).

Combining equation 12 and the last inequality, we arrive at

$$\begin{aligned}
\mathbb{E}[f(z^{k+1})] \leq{} & \mathbb{E}[f(z^k)] + \alpha \frac{\rho_0}{2}L^2\mathbb{E}[\|z^k - x^k\|^2] \\
& + (\alpha \frac{1}{2\rho_0} - \alpha)\mathbb{E}[\|g^k\|^2] + \frac{L\alpha^2}{2}\mathbb{E}[\|\tilde{g}^k\|^2].
\end{aligned}$$

By construction, $z_j^k - x_j^k = -\frac{\beta}{1-\beta}\alpha\tilde{m}_j^{k-1}$ for $j \in J_k$, 0 otherwise. Consequently,

$$\mathbb{E}[f(z^{k+1})] \leq \mathbb{E}[f(z^k)] + \alpha^3\frac{\rho_0}{2}L^2(\frac{\beta}{1-\beta})^2\mathbb{E}\left[\sum_{j \in J_k}\|\tilde{m}_j^{k-1}\|^2\right]$$

$$+ (\alpha\frac{1}{2\rho_0} - \alpha)\mathbb{E}[\|g^k\|^2] + \frac{L\alpha^2}{2}\mathbb{E}[\|\tilde{g}^k\|^2]. \tag{13}$$

Let $k_j = k - t_j^{k-1} + 1$. Then, from Lemma E.2 we know that

$$\mathbb{E}\left[\sum_{j \in J_k}\|\tilde{m}_j^{k-1}\|^2\right] \leq 2\mathbb{E}\left[\sum_{j \in J_k}\|\tilde{m}_j^{k-1} - m_j^{k-1}\|^2\right] + 2\mathbb{E}\left[\sum_{j \in J_k}\|m_j^{k-1}\|^2\right]$$

$$\leq 2\frac{1-\beta}{1+\beta}\mathbb{E}\left[\sum_{j \in J_k}\sigma_j^2 + 2\sum_{j \in J_k}\|m_j^{k-1}\|^2\right]$$

$$\mathbb{E}\left[\sum_{j \in J_k}\|m_j^{k-1}\|^2\right] = \mathbb{E}\left[\sum_{j \in J_k}(1 - \beta^{(k-1)_j})^2\left\|\frac{m_j^{k-1}}{(1 - \beta^{(k-1)_j})}\right\|^2\right] \tag{14}$$

$$\leq 2\mathbb{E}\left[\sum_{j \in J_k}(1 - \beta^{(k-1)_j})^2\left\|\frac{m_j^{k-1}}{(1 - \beta^{(k-1)_j})} - g_j^k\right\|^2\right] + 2\mathbb{E}\left[\sum_{j \in J_k}\|g_j^k\|^2\right]$$

$$\mathbb{E}\left[\|\tilde{g}^k\|^2\right] \leq \sigma^2 + \mathbb{E}[\|g^k\|^2].$$

Putting these into equation 13, we arrive at

$$\mathbb{E}[f(z^{k+1})] \leq \mathbb{E}[f(z^k)] + \left(-\alpha + \alpha\frac{1}{2\rho_0} + 2\alpha^3\rho_0L^2\left(\frac{\beta}{1-\beta}\right)^2 + \frac{L\alpha^2}{2}\right)\mathbb{E}[\|g^k\|^2]$$

$$+ \left(\alpha^3\rho_0L^2\left(\frac{\beta}{1-\beta}\right)^2\frac{1-\beta}{1+\beta}\sigma^2 + \frac{L\alpha^2}{2}\sigma^2\right)$$

$$+ 2\alpha^3\rho_0L^2\left(\frac{\beta}{1-\beta}\right)^2\mathbb{E}\left[\sum_{j \in J_k}(1 - \beta^{(k-1)_j})^2\left\|\frac{m_j^{k-1}}{(1 - \beta^{(k-1)_j})} - g_j^k\right\|^2\right].$$

Notice that if $j \in J^k$, then $(k-1)_j = k_j - 1$. Therefore,

$$\mathbb{E}\left[\left\|\frac{m_j^k}{(1 - \beta^{k_j})} - g_j^k\right\|^2\right] = \mathbb{E}\left[\left\|\frac{\beta m_j^{k-1} + (1-\beta)g_j^k}{(1 - \beta^{k_j})} - g_j^k\right\|^2\right]$$

$$= \beta^2\mathbb{E}\left[\left(\frac{(1 - \beta^{k_j-1})}{(1 - \beta^{k_j})}\right)^2\left\|\frac{m_j^{k-1}}{(1 - \beta^{(k-1)_j})} - g_j^k\right\|^2\right].$$

Substituting the above into the last inequality produces

$$\mathbb{E}[f(z^{k+1})] \leq \mathbb{E}[f(z^k)] + \left(-\alpha + \alpha\frac{1}{2\rho_0} + 2\alpha^3\rho_0L^2(\frac{\beta}{1-\beta})^2 + \frac{L\alpha^2}{2}\right)\mathbb{E}[\|g^k\|^2]$$

$$+ \left(\alpha^3\rho_0L^2(\frac{\beta}{1-\beta})^2\frac{1-\beta}{1+\beta}\sigma^2 + \frac{L\alpha^2}{2}\sigma^2\right) \tag{15}$$

$$+ 2\alpha^3\rho_0L^2\left(\frac{1}{1-\beta}\right)^2\mathbb{E}\left[\sum_{j \in J_k}(1 - \beta^{k_j})^2\left\|\frac{m_j^k}{(1 - \beta^{k_j})} - g_j^k\right\|^2\right].$$

Finally, $\rho_0 = \frac{1-\beta}{2L\alpha}$ gives

$$\mathbb{E}[f(z^{k+1})] \leq \mathbb{E}[f(z^k)] + \left(-\alpha + \frac{1+\beta^2}{1-\beta}L\alpha^2 + \frac{1}{2}L\alpha^2\right)\mathbb{E}[\|g^k\|^2]$$
$$+ \left(\frac{\beta^2}{2(1+\beta)} + \frac{1}{2}\right)L\alpha^2\sigma^2 + \frac{L\alpha^2}{1-\beta}\mathbb{E}\left[\sum_{j\in J_k}(1-\beta^{k_j})^2\left\|\frac{m_j^k}{(1-\beta^{k_j})} - g_j^k\right\|^2\right].$$

$\square$

### E.1. Convergence of Algorithm 3

Firstly, by combining results from prior section, we can bound our Lyapunov function $L^k$ defined in equation 9.

**Proposition E.6.** *Let Assumption 5.1 hold and $\alpha \leq \frac{1-\beta}{2\sqrt{2}L\sqrt{p_{\max}^k}\sqrt{\beta+\beta^2}}$ in Algorithm 3. Let $\{c_i\}_{i=1}^{\infty}$ in equation 9 be defined by*

$$c_1 = \frac{\frac{\beta+\beta^2}{(1-\beta)^3}L^3\alpha^2}{1 - 4\alpha^2\frac{\beta+\beta^2}{(1-\beta)^2}L^2}, \qquad c_{i+1} = c_i - \left(4c_1\alpha^2 + \frac{L\alpha^2}{1-\beta}\right)\beta^i(i + \frac{\beta}{1-\beta})L^2 \quad \text{for all } i \geq 1.$$

*Then, $c_i > 0$ for all $i \geq 1$, and*

$$\mathbb{E}[L^{k+1} - L^k] \leq \left(-\alpha + \frac{3-\beta+\beta^2}{2(1-\beta)}L\alpha^2 + 4c_1\alpha^2\right)\mathbb{E}[\|g^k\|^2] \tag{16}$$
$$+ \left(\frac{\beta^2}{2(1+\beta)}L\alpha^2\sigma^2 + \frac{1}{2}L\alpha^2\sigma^2 + 2c_1\alpha^2\sigma^2\right).$$

*Proof.* Recall that $L^k$ is defined as

$$L^k = f(z^k) - f^* + \sum_{i=1}^{k-1}c_i\|x^{k+1-i} - x^{k-i}\|^2,$$

Therefore, by equation 15 we know that

$$\mathbb{E}[L^{k+1} - L^k] \leq$$
$$(-\alpha + \frac{1+\beta^2}{1-\beta}L\alpha^2 + \frac{1}{2}L\alpha^2)\mathbb{E}[\|g^k\|^2]$$
$$+ \sum_{i=1}^{k-1}(c_{i+1} - c_i)\mathbb{E}[\|x^{k+1-i} - x^{k-i}\|^2] + c_1\mathbb{E}[\|x^{k+1} - x^k\|^2] \tag{17}$$
$$+ \left(\frac{\beta^2}{2(1+\beta)} + \frac{1}{2}\right)L\alpha^2\sigma^2 + \frac{L\alpha^2}{1-\beta}\mathbb{E}\left[\sum_{j\in J_k}(1-\beta^{k_j})^2\left\|\frac{m_j^k}{(1-\beta^{k_j})} - g_j^k\right\|^2\right].$$

To bound the $c_1\mathbb{E}[\|x^{k+1} - x^k\|^2]$ term, we need the following inequalities, which are obtained similarly as equation 14.

$$\mathbb{E}[\|\tilde{m}^k\|^2] \leq 2\frac{1-\beta}{1+\beta}\sigma^2 + 2\mathbb{E}[\|m^k\|^2]$$
$$\mathbb{E}[\|m^k\|^2] \leq 2\mathbb{E}\left[\sum_{j\in J_k}(1-\beta^{k_j})^2\left\|\frac{m_j^k}{(1-\beta^{k_j})} - g_j^k\right\|^2\right] + 2\mathbb{E}\left[\|g^k\|^2\right] \tag{18}$$
$$\mathbb{E}[\|\tilde{g}^k\|^2] \leq \sigma^2 + \mathbb{E}[\|g^k\|^2].$$

Let $\Pr_{k-1}[j \in J_k] = p_j^k$ and $p_{\min}^k := \min_{j \in [d]}\{p_j^k\}$. Then, $c_1\mathbb{E}[\|x^{k+1} - x^k\|^2]$ can be bounded as

$$c_1\mathbb{E}[\|x^{k+1} - x^k\|^2] = c_1\alpha^2\mathbb{E}[\|\tilde{u}^k\|^2] = c_1\alpha^2\mathbb{E}\left[\sum_{j \in J_k}\|\tilde{m}_j^k\|^2 + \sum_{j \notin J_k}\|\tilde{g}_j^k\|^2\right]$$

$$\leq c_1\alpha^2\mathbb{E}\left[\|\tilde{m}^k\|^2 + (1 - p_{\min}^k)\|\tilde{g}^k\|^2\right]$$

$$\leq c_1\alpha^2\left(\left(2\frac{1-\beta}{1+\beta} + 1 - p_{\min}^k\right)\sigma^2 + 5\mathbb{E}[\|g^k\|^2]\right)$$

$$+ 4c_1\alpha^2\mathbb{E}\left[\sum_{j \in J_k}(1 - \beta^{k_j})^2\left\|\frac{m_j^k}{(1 - \beta^{k_j})} - g_j^k\right\|^2\right]$$

Combine this with equation 17, we obtain

$$\mathbb{E}[L^{k+1} - L^k]$$

$$\leq \left(-\alpha + \frac{1 + \beta^2}{1 - \beta}L\alpha^2 + \frac{1}{2}L\alpha^2 + 5c_1\alpha^2\right)\mathbb{E}[\|g^k\|^2] + \left(\frac{\beta^2}{2(1 + \beta)} + \frac{1}{2} + \frac{c_1}{L}\left(2\frac{1-\beta}{1+\beta} + 1 - p_{\min}^k\right)\right)L\alpha^2\sigma^2$$

$$+ \sum_{i=1}^{k-1}(c_{i+1} - c_i)\mathbb{E}[\|x^{k+1-i} - x^{k-i}\|^2] \tag{19}$$

$$+ \left(4c_1\alpha^2 + \frac{L\alpha^2}{1 - \beta}\right)\mathbb{E}\left[\sum_{j \in J_k}(1 - \beta^{k_j})^2\left\|\frac{m_j^k}{(1 - \beta^{k_j})} - g_j^k\right\|^2\right].$$

In the rest of the proof, let us show that the sum of the last two terms in equation 19 is non-positive.

First of all, by Lemma E.3 we know that

$$\mathbb{E}\left[\sum_{j \in J_k}(1 - \beta^{k_j})^2\left\|\frac{m_j^k}{(1 - \beta^{k_j})} - g_j^k\right\|^2\right] \leq \mathbb{E}\left[p_{\max}^k\sum_{i=1}^{k-1}a_{k,i}\|x^{i+1} - x^i\|^2\right],$$

where

$$a_{k,i} = L^2\beta^{k-i}\left(k - i + \frac{\beta}{1 - \beta}\right).$$

Or equivalently,

$$\mathbb{E}\left[\sum_{j \in J_k}(1 - \beta^{k_j})^2\left\|\frac{m_j^k}{(1 - \beta^{k_j})} - g_j^k\right\|^2\right] \leq \mathbb{E}\left[\sum_{i=1}^{k-1}p_{\max}^k a_{k,k-i}\|x^{k+1-i} - x^{k-i}\|^2\right],$$

where

$$a_{k,k-i} = L^2\beta^i\left(i + \frac{\beta}{1 - \beta}\right).$$

Therefore, to make the sum of the last two terms of equation 19 to be non-positive, we need to have

$$c_{i+1} \leq c_i - \left(4c_1\alpha^2 + \frac{L\alpha^2}{1 - \beta}\right)L^2p_{\max}^i\beta^i\left(i + \frac{\beta}{1 - \beta}\right)$$

for all $i \geq 1$. To satisfy this inequality, we choose

$$c_{i+1} = c_i - \left(4c_1\alpha^2 + \frac{L\alpha^2}{1 - \beta}\right)L^2\beta^i p_{\max}^i\left(i + \frac{\beta}{1 - \beta}\right)$$

for all $i \geq 1$, which implies that

$$c_i = c_1 - \left(4c_1\alpha^2 + \frac{L\alpha^2}{1-\beta}\right) L^2 \sum_{l=1}^{i-1} \beta^i p_{\max}^i \left(i + \frac{\beta}{1-\beta}\right).$$

To have $c_i > 0$ for all $i \geq 1$, we can set $c_1$ as

$$c_1 = \left(4c_1\alpha^2 + \frac{L\alpha^2}{1-\beta}\right) L^2 \hat{p}_{\max}^k \sum_{i=1}^{\infty} \beta^i \left(i + \frac{\beta}{1-\beta}\right).$$

where, $\hat{p}_{\max}^k = \max_{i \in [k]} \{p_{\max}^i\}$. Since

$$\sum_{i=1}^{j} i\beta^i = \frac{1}{1-\beta} \left(\frac{\beta(1-\beta^j)}{1-\beta} - j\beta^{j+1}\right),$$

we have $\sum_{i=1}^{\infty} i\beta^i = \frac{\beta}{(1-\beta)^2}$ and

$$c_1 = \left(4c_1\alpha^2 + \frac{L\alpha^2}{1-\beta}\right) L^2 \hat{p}_{\max}^k \frac{\beta + \beta^2}{(1-\beta)^2},$$

which implies that

$$c_1 = \frac{\alpha^2 L^3 \hat{p}_{\max}^k \frac{\beta+\beta^2}{(1-\beta)^3}}{1 - 4\alpha^2 \frac{\beta+\beta^2}{(1-\beta)^2} \hat{p}_{\max}^k L^2}. \tag{20}$$

Notice that $\alpha \leq \frac{1-\beta}{2\sqrt{2}L\sqrt{\hat{p}_{\max}^k}\sqrt{\beta+\beta^2}}$ ensures $c_1 > 0$.

Therefore,

$$\mathbb{E}[L^{k+1} - L^k] \leq \left(-\alpha + \frac{3-\beta+2\beta^2}{2(1-\beta)} L\alpha^2 + 5c_1\alpha^2\right) \mathbb{E}[\|g^k\|^2]$$
$$+ \left(\frac{\beta^2}{2(1+\beta)} L\alpha^2\sigma^2 + \frac{1}{2} L\alpha^2\sigma^2 + c_1\alpha^2\sigma^2 \left(2\frac{1-\beta}{1+\beta} + 1 - p_{\min}^k\right)\right).$$

□

By telescoping equation 16, we obtain the convergence bound of our proposed algorithm under nonconvex settings.

**Theorem E.7.** *Let Assumption 5.1 hold and $\alpha^k = \alpha \leq \frac{1-\beta}{L(4-\beta+\beta^2)}$. Then, the iterates of Algorithm 3 satisfy*

$$\frac{1}{k} \sum_{i=1}^{k} \mathbb{E}[\|g^i\|^2] \leq \mathcal{O}\left(\frac{f(x^1) - f^*}{k\alpha} + L\alpha\sigma^2 \left(1 + \frac{\hat{p}_{\max}^k(1 - \bar{p}_{\min}^k)\beta}{(1-\beta)}\right)\right), \tag{21}$$

*where $\bar{p}_{\min}^k = \frac{1}{k}\sum_{i=1}^{k} \bar{p}_{\min}^i$ and $\hat{p}_{\max}^k = \max_{i \in [k]}\{p_{\max}^i\}$.*

*Proof.* From equation 16 we know that

$$\mathbb{E}[L^{k+1} - L^k] \leq -R_1\mathbb{E}[\|g^k\|^2] + R_2^k, \tag{22}$$

where

$$R_1 = -\alpha + \frac{3 - \beta + \beta^2}{2(1 - \beta)} L\alpha^2 + 4c_1\alpha^2,$$

$$R_2 = \frac{\beta^2}{2(1 + \beta)} L\alpha^2\sigma^2 + \frac{1}{2} L\alpha^2\sigma^2 + c_1\alpha^2\sigma^2 \left( 2\frac{1 - \beta}{1 + \beta} + 1 - p_{\min}^k \right).$$

We further define

$$\bar{R}_2 = \frac{\beta^2}{2(1 + \beta)} L\alpha^2\sigma^2 + \frac{1}{2} L\alpha^2\sigma^2 + c_1\alpha^2\sigma^2 \left( 2\frac{1 - \beta}{1 + \beta} + 1 - \bar{p}_{\min}^k \right),$$

where $\bar{p}_{\min}^k = \frac{1}{k}\sum_{i=1}^k \bar{p}_{\min}^i$.

Telescoping equation 22 yields

$$L^1 \geq \mathbb{E}[L^1 - L^{k+1}] \geq R_1 \sum_{i=1}^k \mathbb{E}[\|g^i\|^2] - \sum_{k=1}^k R_2^k,$$

and therefore

$$\frac{1}{k}\sum_{i=1}^k \mathbb{E}[\|g^i\|^2] \leq \frac{L^1}{kR_1} + \frac{\bar{R}_2}{R_1}. \tag{23}$$

In the rest of the proof, we will appropriately bound $R_1$ and $\bar{R}_2$.

First, let us show that $R_1 \geq \frac{\alpha}{2}$ and $\alpha \leq \min\left\{ \frac{1-\beta}{L(4-\beta+\beta^2)}, \frac{1-\beta}{2\sqrt{2}L\sqrt{\hat{p}_{\max}^k}\sqrt{\beta+\beta^2}} \right\}$.

From equation 20 we know that

$$c_1 = \frac{\alpha^2 L^3 \hat{p}_{\max}^k \frac{\beta+\beta^2}{(1-\beta)^3}}{1 - 4\alpha^2 \frac{\beta+\beta^2}{(1-\beta)^2} L^2 \hat{p}_{\max}^k}.$$

Since $\alpha \leq \frac{1-\beta}{2\sqrt{2}L\sqrt{\hat{p}_{\max}^k}\sqrt{\beta+\beta^2}}$, we have

$$4\alpha^2 \frac{\beta + \beta^2}{(1 - \beta)^2} L^2 \hat{p}_{\max}^k \leq \frac{1}{2}.$$

Thus,

$$c_1 \leq \alpha^2 L^3 \hat{p}_{\max}^k \frac{\beta + \beta^2}{(1 - \beta)^3} \leq \frac{L}{8(1 - \beta)}.$$

Therefore, in order to ensure $R_1 \geq \frac{\alpha}{2}$, it suffices to have

$$\frac{3 - \beta + \beta^2}{2(1 - \beta)} L\alpha + \frac{\alpha L}{2(1 - \beta)} \leq \frac{1}{2}$$

which is equivalent to our condition $\alpha \leq \frac{1-\beta}{L(4-\beta+\beta^2)}$.

For $\bar{R}_2$, we can upperbound $c_1$ using our condition $\alpha \leq \frac{1-\beta}{L(4-\beta+\beta^2)}$. Thus,

$$c_1 \leq \alpha^2 L^3 \hat{p}_{\max}^k \frac{\beta + \beta^2}{(1 - \beta)^3} \leq \frac{\hat{p}_{\max}^k \beta L}{2(1 - \beta)}.$$

Therefore,

$$
\begin{aligned}
\bar{R}_2 &= \frac{\beta^2}{2(1+\beta)} L\alpha^2\sigma^2 + \frac{1}{2} L\alpha^2\sigma^2 + c_1\alpha^2\sigma^2 \left( 2\frac{1-\beta}{1+\beta} + 1 - \bar{p}_{\min}^k \right) \\
&\leq \frac{\beta^2}{2(1+\beta)} L\alpha^2\sigma^2 + \frac{1}{2} L\alpha^2\sigma^2 + \frac{\hat{p}_{\max}^k \beta L\alpha^2\sigma^2}{(1+\beta)} + L\alpha^2\sigma^2 \hat{p}_{\max}^k(1 - \bar{p}_{\min}^k)\frac{\beta}{1-\beta} \\
&\leq \left( \frac{2\beta^2 + 8\hat{p}_{\max}^k}{2(1+\beta)} + \frac{1}{2} + \frac{\hat{p}_{\max}^k(1 - \bar{p}_{\min}^k)\beta}{8(1-\beta)} \right) L\alpha^2\sigma^2.
\end{aligned}
$$

By putting them all together, we obtain

$$
\begin{aligned}
\frac{1}{k}\sum_{i=1}^{k} \mathbb{E}[\|g^i\|^2] &\leq \frac{2\left(f(x^1) - f^*\right)}{k\alpha} + \left( \frac{2\beta^2 + 8\hat{p}_{\max}^k}{2(1+\beta)} + \frac{1}{2} + \frac{\hat{p}_{\max}^k(1 - \bar{p}_{\min}^k)\beta}{8(1-\beta)} \right) L\alpha\sigma^2 \\
&= \mathcal{O}\left( \frac{f(x^1) - f^*}{k\alpha} + L\alpha\sigma^2 \left( 1 + \frac{\hat{p}_{\max}^k(1 - \bar{p}_{\min}^k)\beta}{(1-\beta)} \right) \right).
\end{aligned}
$$

$\square$

# F. Limitations

We would also like to acknowledge the limitations of this work. Due to computational constraints, we were unable to conduct experiments on pre-training 7B+ LLMs, which is crucial for understanding the potential of our approach when scaling. Furthermore, our experiments are limited to training language models, although memory-efficient optimization could also be beneficial for training diffusion models. Finally, there may be a better method for selecting the next state-full subspace during the training. We leave the exploration of more sophisticated selection strategies for future work.

# G. Simplified algorithms pseudocode

In this section we present the simplified pseudocode of FRUGAL. In Algorithm 4 one can find optimizer steps both for FRUGAL with SVD projection (GaLore-like (Zhao et al., 2024a)) and Block projection (BAdam-like (Luo et al., 2024)).

**Algorithm 4** FRUGAL step pseudocode, PyTorch-like

```python
 1: def svd_or_randk_step(self):
 2:     for param in self.params:
 3:         grad = param.grad
 4:         param_state = self.state[param]
 5:         # update projector if necessary
 6:         if self.step % self.update_gap == 0:
 7:             param_state["projector"] = self.update_proj(grad)
 8:         projector = param_state["projector"]
 9:         # obtain state-full grad and state-free grad
10:         grad_full = projector.proj_down(grad)
11:         grad_free = grad_full - projector.proj_up(grad_full)
12:         # reset state-full optimizer state if necessary
13:         if self.step % self.update_gap == 0:
14:             param_state["exp_avg"] = torch.zeros_like(grad_full)
15:             param_state["exp_avg_sq"] = torch.zeros_like(grad_full)
16:         # state-full subspace update
17:         self.step += 1
18:         update_full = self.state_full_step(grad_full, param_state)
19:         update_full = projector.proj_up(update_full)
20:         # state-free subspace update
21:         update_free = self.state_free_step(grad_free)
22:         # perform resulting update
23:         update = update_full + update_free
24:         param.add_(update)
25:
26: def block_step(self):
27:     # change state-full and state-free blocks if necessary
28:     if self.step % self.update_gap == 0:
29:         indices_full = self.update_indices(indices_full)
30:         for idx, param in enumerate(self.params):
31:             grad = param.grad
32:             param_state = self.state[param]
33:             if idx in indices_full:
34:                 # reset state-full optimizer state
35:                 param_state["exp_avg"] = torch.zeros_like(grad)
36:                 param_state["exp_avg_sq"] = torch.zeros_like(grad)
37:                 param_state["full_subspace"] = True
38:             else:
39:                 # free state-full optimizer state to save memory
40:                 param_state.clear()
41:                 param_state["full_subspace"] = False
42:     # perform updates
43:     for param in self.params:
44:         grad = param.grad
45:         param_state = self.state[param]
46:         # choose the optimizer depending on the block type
47:         if param_state["full_subspace"]:
48:             update = self.state_full_step(grad, param_state)
49:         else:
50:             update = self.state_free_step(grad)
51:         # perform resulting update
52:         param.add_(update)
```

---

**Algorithm 5** Examples of state-full and state-free steps for Algorithm 4

```
 1: def state_full_adam_step(self, grad, param_state):
 2:     exp_avg = param_state["exp_avg"]
 3:     exp_avg_sq = param_state["exp_avg_sq"]
 4:     step = self.step
 5:     beta1, beta2 = self.betas
 6:     exp_avg.mul_(beta1).add_(grad, alpha=1.0-beta1)
 7:     exp_avg_sq.mul_(beta2).addcmul_(grad, grad, value=1.0-beta2)
 8:     denom = exp_avg_sq.sqrt()
 9:     step_size = self.lr_full
10:     if self.correct_bias:
11:         bias_correction1 = 1.0 - beta1 ** step
12:         bias_correction2 = 1.0 - beta2 ** step
13:         step_size = self.lr_full / bias_correction1
14:         bias_correction2_sqrt = math.sqrt(bias_correction2)
15:         denom.div_(bias_correction2_sqrt)
16:     denom.add_(self.eps)
17:     update_full = exp_avg / denom * (-step_size)
18:     return update_full
19:
20: def state_free_signsgd_step(self, grad):
21:     update_free = -self.lr_free * grad.sign()
22:     return update_free
```

---

