# OpenReview forum: "FRUGAL: Memory-Efficient Optimization by Reducing State Overhead for Scalable Training"
_ICML.cc/2025/Conference — ICML 2025 poster_

### Official Review · Reviewer_3FWu · 2025-03-10

**Overall Recommendation:** 2

**Summary:**

This paper presents a novel approach to reduce memory overhead during LLM training by dividing the model parameters into two distinct groups. One group is optimized using Adam-family optimizers, which maintain optimizer states, while the other group is trained with state-free optimization methods such as SGD and sign-SGD. This design aims to minimize memory consumption while maintaining acceptable performance degradation.

**Claims And Evidence:**

The claim that this work is "the first to train the majority of language model parameters using signSGD without momentum" appears to be overstated. Recent advancements, such as the Lion optimizer, exhibit a highly similar approach that closely aligns with this claim. Moreover, with carefully chosen hyperparameters, the claim (using signSGD for LLM) may effectively reduce to a special case of Lion, further diminishing the novelty of the stated contribution.

**Essential References Not Discussed:**

Not Found.

**Experimental Designs Or Analyses:**

The experimental results provided for fine-tuning are insufficient. Notably, the fine-tuning experiments on RoBERTa using the GLUE benchmark are somewhat outdated in the context of modern LLM benchmarks. Consequently, the conclusions drawn from these experiments may have limited generalizability to contemporary LLM training and fine-tuning settings.

**Methods And Evaluation Criteria:**

- Algorithm 2 seems problematic in its treatment of parameters that are not part of the set $J_k$. Specifically, the algorithm updates
$m_j^k$  for these parameters, which equivalently assumes their gradients are zero. If the set $J_k$  is sampled randomly, Algorithm 2 would quite resemble standard SGD. Furthermore, as $\beta$ approaches 1, $m_j^k$ converges to zero, preventing the algorithm from achieving convergence. This issue raises concerns about the proposed method's stability and effectiveness.

- The purpose and functionality of Line 6 in Algorithm 1 are unclear. This step fails to effectively distinguish between state-full and state-free parameters, diminishing the clarity of the proposed method's design.

**Other Comments Or Suggestions:**

See above

**Other Strengths And Weaknesses:**

See above

**Questions For Authors:**

An intriguing observation arises from Table 1, where the block-wise results underperform compared to the SVD-based method. Since SVD can be viewed as a special case of block-wise projection (i.e., full-rank projection without dimensionality reduction), it is unexpected that the block-wise approach yields inferior results. A deeper investigation into this discrepancy would enhance the paper's insights.

**Relation To Broader Scientific Literature:**

N/A

**Theoretical Claims:**

- Theoretical findings, particularly Theorem 5.2, suggest that combining SGD with SGDM results in a consistently worse upper bound. This outcome appears to undermine the theoretical soundness of the proposed FRUGAL method. A more comprehensive discussion or additional theoretical justification is necessary to clarify this apparent limitation.

---

> ### Author Rebuttal · Authors · 2025-04-01
>
> We thank the reviewer for their detailed feedback and respond to their concerns and questions hereafter.
>
> The tables that we will refer to using an apostrophe (e.g., Table 1') can be found at the anonymous link https://anonymous.4open.science/r/frugal-618F/rebuttal.pdf.
>
> >Recent advancements, such as the Lion optimizer, exhibit a highly similar approach that closely aligns with this claim.
>
> We kindly disagree with the reviewer that the existence of the Lion work diminishes the contribution of our research. While some prior works [1,2] have indeed used sign-based approaches, it is important to note that all these methods **incorporated momentum**. To the best of our knowledge, we are the first to use signSGD **without momentum** to successfully train the majority of language model parameters.
>
> >Algorithm 2 seems problematic in its treatment of parameters that are not part of the set $J_k$. Specifically, the algorithm updates $m_j^k$ for these parameters, which equivalently assumes their gradients are zero.
>
> We thank the reviewer for pointing out this minor inaccuracy with the update of $m$ in line 3. We intended the following update formula:
>
> $\tilde{m}_j^{k} \leftarrow \begin{cases}  (1-\beta)\tilde{g}_j^{k} + \beta\tilde{m}_j^{k-1} & \text{if } j \in J_k; \\
> 0 & \text{otherwise;}
> \end{cases}$
>
> This ensures that when $j$ re-enters $J_k$, the value $m_j$ is reset to 0 (this is assumed in Equation 3).
>
> >If the set $J_k$ is sampled randomly, Algorithm 2 would quite resemble standard SGD. Furthermore, as $\beta$ approaches 1, $m_j^k$ converges to zero, preventing the algorithm from achieving convergence.
>
> We kindly disagree with the reviewer on this issue. Since $j\notin J_k$ are updated without using $\beta$, and for $j\in J_k$, the algorithm implements standard SGDM, which works well with $\beta$ values close to 1, we do not see why this should prevent the convergence of our algorithm. Moreover, this statement contradicts Theorem 5.2, which we consider to be correct.
>
> >The purpose and functionality of Line 6 in Algorithm 1 are unclear. This step fails to effectively distinguish between state-full and state-free parameters, diminishing the clarity of the proposed method's design.
>
> By $P^{-1}$ in Line 6, we mean the right inverse of $P$. Thus, $P^{-1}(P(g))$ represents the projection of $g$ onto the low-rank subspace. We will add this clarification in the final version of the paper.
>
> >Theoretical findings, particularly Theorem 5.2, suggest that combining SGD with SGDM results in a consistently worse upper bound.
>
> We note that this is the case only in deterministic scenario, where the main issue is that the bias from momentum affects all the coordinates. At the same time, the variance reduction effect is only present in momentum coordinates. This is not improvable in the worst case, but in the average (best) case, the benefit of momentum can be more prevalent compared to (sign)SGD, which explains our numerical findings.
>
> >The experimental results provided for fine-tuning are insufficient. Notably, the fine-tuning experiments on RoBERTa using the GLUE benchmark are somewhat outdated in the context of modern LLM benchmarks.
>
> At the reviewer's request, we conducted additional fine-tuning experiments. Specifically, we evaluated the effectiveness of various fine-tuning algorithms on LLaMA 3.1-8B in Commonsense Reasoning.
>
> Following the setup in [3], we fine-tuned the model on the Commonsense170K dataset [3] and evaluated accuracy across 8 datasets (see the full list in [3], Section 3.1). We used the same hyperparameters as in [3], and varied the learning rate among [5e-6, 1e-5, 2e-5, 5e-5, 1e-4, 2e-4] for all methods.
>
> The results, presented in Table 6', show that FRUGAL slightly outperforms both LoRA and GaLore. Notably, it achieves this even with $\rho=0.0$ (using 0 memory for optimizer state).
>
> >Since SVD can be viewed as a special case of block-wise projection (i.e., full-rank projection without dimensionality reduction), it is unexpected that the block-wise approach yields inferior results.
>
> We believe there has been a slight misunderstanding here regarding the SVD and Blockwise methods of selecting the state-full subspace. By SVD, we meant the approach of applying a low-rank projection to **each** trainable matrix. By block-wise, we refer to a block coordinate descent-like strategy where a subset of matrices resides  in the state-full subspace, while **the others entirely remain in the state-free subspace**.
>
> To sum up, we believe that we addressed all the reviewer's concerns and questions, none of which is a serious issue with our approach. Therefore, we would kindly request the reviewer to reconsider their score.
>
> [1] Chen et al., Symbolic discovery of optimization algorithms, NeurIPS 2023.
>
> [2] Zhao et al., Deconstructing what makes a good optimizer for language models, ICLR 2025.
>
> [3] Hu et al., Llm-adapters: An adapter family for parameter-efficient fine-tuning of large language models, EMNLP 2023.

---

### Official Review · Reviewer_Hho9 · 2025-03-12

**Overall Recommendation:** 3

**Summary:**

This paper focuses on memory-efficient training by using different optimizers on different subspaces of the gradient, and uses different methods for projecting the gradient onto the state-full subspace. They extend AdaLayer to use signSGD instead of SGDM.

**Claims And Evidence:**

The proposed method outperforms existing methods such as GaLore and BAdam when training Llama-60M, 130M, 350M, and 1B on the C4 dataset.

**Essential References Not Discussed:**

The essential reference [Zhao et al. 2024b] is cited, but is not emphasized properly.

**Experimental Designs Or Analyses:**

The paper combines different lineages of memory-efficient optimizers 1) low-rank projection-based, 2) block coordinate descent, 3) sign-based, and 4) state-free/full hybrid, but this makes it difficult to see where the efficiency is actually coming from. Adding "AdamW, ρ=0.25" and "AdaLayer [Zhao et al. 2024b] ρ=0" to Table 2 would help quantify the contribution to memory-efficiency of each method. It would also be nice to have some comments about how the memory-efficiency improves with scale (number of parameters).

**Methods And Evaluation Criteria:**

The authors perform extensive experiments across various model scales and hyperparameter settings, providing a thorough analysis of optimizer performance and stability. The benchmark datasets are standard and I did not find anything missing in the evaluation criteria.

**Other Comments Or Suggestions:**

I am particularly interested in the extension of AdaLayer's state-free optimizer from SGDM to signSGD. It would be nice to see an ablation for just this change. I would also like to see experiments for 0<ρ<0.25 to see whether it yields similar accuracy while reducing the memory consumption.

**Other Strengths And Weaknesses:**

In the paper, it says "ρ denotes the proportion of the Linear layer parameters in the state-full subspace", but it is unclear how the authors choose which linear layers to optimize with the state-full optimizer. Depending on this choice, perhaps "FRUGAL, ρ=0.25" could yield even better results?

**Questions For Authors:**

State-free optimizers are more sensitive to hyperparameters, so adapting them in a certain sub-space may inherit this weakness. Have the authors encountered such problems?

**Relation To Broader Scientific Literature:**

Although the paper puts a lot of emphasis on optimizers such as GaLore, ReLoRA, and BAdam, the key contribution of this paper is more related to AdaLayer [Zhao et al. 2024b], but with signSGD instead of SGDM. If a majority of layers can be optimized with state-free optimizers, the efficiency of the state-full part is less significant.

**Theoretical Claims:**

This work presents the first theoretical analysis of an extended block coordinate descent framework where the remaining layers are also updated with a different algorithm. I did not find any errors in the proofs.

---

> ### Author Rebuttal · Authors · 2025-04-01
>
> We appreciate the reviewer's comprehensive feedback. We are glad that they appreciated the theoretical convergence guarantees and strong experimental results. We also answer their questions below. The tables that we will refer to using an apostrophe (e.g., Table 1') can be found at the anonymous link https://anonymous.4open.science/r/frugal-618F/rebuttal.pdf.
>
> >The paper combines different lineages of memory-efficient optimizers …  but this makes it difficult to see where the efficiency is actually coming from.
>
> We agree that understanding how different components of the overall framework affect the final results is critically important. Below, we address each of the enumerated points:
>
> >1) low-rank projection-based, 2) block coordinate descent,
>
> Within our framework, we consider two main options for selecting the state-full subspace: 1. a low-rank projection for each trainable matrix (denoted as SVD in the paper), and 2. a block coordinate descent-like approach where some subset of matrices is fully within the state-full subspace while all others are in the state-free subspace (denoted as Blockwise). Experimental results comparing these approaches are presented in Table 1. Also, see the detailed discussion of memory and computational requirements in Section 4 and Appendix B.
>
> >3) sign-based
>
> As state-free optimizers, we only considered signSGD and SGD. The discussion and experimental results can be found in Section 4 and Table 8.
>
> >4) state-free/full hybrid
>
> The incorporation of state-free subspace optimization is the main contribution of our framework. The advantages of this approach are demonstrated, for example, in Tables 1 and 2.
>
> To summarize the results, the most significant impact on the metrics is due to the addition of the state-full/state-free hybrid, followed by the use of sign-based optimization for the state-free subspace. The type of projection has the least effect.
>
> >Adding "AdamW, ρ=0.25" to Table 2 would help quantify the contribution to memory-efficiency of each method.
>
> We assume that by "AdamW, $\rho=0.25$" the reviewer means that only $\rho=0.25$ of the Linear layer parameters are unfrozen and trained using AdamW. We would like to note that this setup is very similar to our baseline BAdam, where the set of unfrozen parameters changes every $T$ steps. For a more complete picture, we also conducted experiments with a setup where the active parameters do not change throughout the training process. As active parameters, we experimented with selecting the first and last $\rho=0.25$ layers. The results are presented in Table 3'.
>
> >I am particularly interested in the extension of AdaLayer's state-free optimizer from SGDM to signSGD. It would be nice to see an ablation for just this change. I would also like to see experiments for 0<ρ<0.25.
>
> Since the original version of AdaLayer (Algorithm 1 from [1]) does not use SGDM, we assume that the reviewer was referring to its version in Section 3.2 [1].
> As requested, we conducted pretraining experiments on models up to 350M, replacing SGDM with signSGD in this version of the algorithm and also using FRUGAL with AdaLayer (see implementation here https://anonymous.4open.science/r/frugal-618F/adalayer.py) as the state-full optimizer with different values of $\rho$. The final results, which also include the results from Table 9 of the original paper, are presented in Table 4'. As can be seen, the variant with AdaLayer demonstrates a very similar trend to the original FRUGAL, albeit with slightly worse values.
>
> >The essential reference [Zhao et al. 2024b] is cited, but is not emphasized properly.
>
> While the AdaLayer shares some similarities with FRUGAL with ρ=0.0, we consider our framework more general since *it allows configurations at other values of $\rho$ that have no equivalent in AdaLayer•. Therefore, we believe our work is still closer to our primary baselines, such as GaLore.
>
> >It would also be nice to have some comments about how the memory-efficiency improves with scale (number of parameters).
>
> An exact estimate of the memory savings from using FRUGAL compared to AdamW is provided in Table 5'.
>
> > It is unclear how the authors choose which linear layers to optimize with the state-full optimizer. Depending on this choice, perhaps "FRUGAL, ρ=0.25" could yield even better results?
>
> We compared several methods of alternating between state-full subspaces in Section 6.4 and Table 11. The optimal selection of stateful subspaces is a complex task, and we leave it for future work.
>
> >State-free optimizers are more sensitive to hyperparameters, so adapting them in a certain sub-space may inherit this weakness. Have the authors encountered such problems?
>
> We did not observe significant sensitivity in our experiments. For example, varying $\beta_2$, as presented in Table 1', affects FRUGAL in approximately the same way as it does AdamW.
>
> [1] Zhao et al., Deconstructing what makes a good optimizer for language models, 2024.

---

> > ### Comment · Reviewer_Hho9 · 2025-04-04
> >
> > Thank you so much for answering all my questions in the review. I think this is valuable work that definitely has merit. Most of the answers addressed my concerns, and I only have one more request before considering to increase my score. In your rebuttal you mention that "the most significant impact on the metrics is due to the addition of the state-full/state-free hybrid". This reinforces my belief that the existing work on state-full/state-free hybrid [Zhao et al. 2024b] should be emphasized a bit more. To readers of this paper it just seems a little confusing when the main contribution in the experiments is coming from state-full/state-free hybrid, but the method part focuses so much on GaLore and so little on AdaLayer. (I declare that I have no relation to the authors of AdaLayer. This is coming from a purely objective perspective of a third party)

---

> > > ### Author Response · Authors · 2025-04-04
> > >
> > > We are glad that we were able to address most of the reviewer's concerns.
> > >
> > > Regarding the reviewer's last question: in our work, we initially focused more on GaLore and BAdam because all variants of Adalayer described in [Zhao et al. 2024] still *maintain a momentum buffer*. Thus, for Adalayer to work, each matrix of size $m\times n$ still requires at least $m\cdot n$ additional memory. Therefore, this algorithm does not fall into the same category of memory-efficient approaches as FRUGAL, GaLore, BAdam, and ReLoRA, which only requires $2\rho \cdot (m \cdot n)$ additional memory, and $\rho$ can be much less than $1/2$. Thus, Adalayer can be considered a 'preconditioner-free' hybrid rather than a 'state-free' one.
> > >
> > > However, we agree that the reviewer's argument is valid. FRUGAL indeed stands out among the baselines precisely because of its hybrid structure. Hybridity also forms the basis of the main theoretical contribution of our work. Therefore, we will definitely add a discussion on hybrid optimizers in Sections 1, 2, and 4 in the camera-ready version of the paper, where we will pay special attention to Adalayer as the closest algorithm in this regard.
> > >
> > > [1] Zhao et al., Deconstructing what makes a good optimizer for language models, ICLR 2025.

---

### Official Review · Reviewer_wtSQ · 2025-03-14

**Overall Recommendation:** 3

**Summary:**

The paper proposes a memory efficient way of combining existing stateful (like Adam) and stateless (like signSGD) optimizers, by running stateful optimizers in a low dimensional space and stateless optimizers in the complementary space. They provide results on pretraining as well as finetuning setup and show that the method outperforms other low memory counterparts such as LORA and BAdam, while being comparable to Adam in some setups.

**Claims And Evidence:**

The paper provides **mostly** adequate evidence for the claims made about the superiority of the FRUGAL optimizer over existing low memory optimizers. I have mentioned mostly as the hyperparameter used for Adam as mentioned in the Appendix has beta 2 of 0.999. Although this has been the default beta2 value in vision literature, in recent LLM works such as Zhao, the optimal beta2 is close to 0.95.

**Essential References Not Discussed:**

One of the crucial references not discussed is AdaMem. It also has the similar idea of decomposing the current update into a ‘top’ direction where momentum is maintained and a bottom direction where only preconditioning happens without any momentum. This is an essential baseline to be compared with.

Vyas et al. 2024 - AdaMeM: Memory Efficient Momentum for Adafactor

**Experimental Designs Or Analyses:**

I looked at the hyperparameter sweeps for most of the experiments. My main concern is the use of beta2 = 0.999 for Adam, which is not standard in language model setups.

**Methods And Evaluation Criteria:**

Yes methods and evaluation criteria make sense.

**Other Comments Or Suggestions:**

The theory provided within AdaMem work clearly shows an improvement based on how momentum is maintained, while in this case, the theory is in a setting where the rates for SGD and SGDM are the same. It would be better to consider a setting where the rates with momentum differ and show how the given optimizer interpolates between the two rates.

**Other Strengths And Weaknesses:**

One of the main strength of the paper is the extensive evaluation of the FRUGAL optimizer on various pretraining and fine tuning settings, and also various ablations showing the effect of various choices on the performance.

The main weakness is the missing reference to AdaMem and a comparison to the optimizer.

**Questions For Authors:**

1. Would it be possible to rerun some of the pretraining experiments with beta2 = 0.95? (even at scale of 130m works)

2. Would it be possible to include AdaMem as a baseline?

3. How does the proposed method differ from AdaMem?

4. Can the gains of the given method be theoretically shown in a similar setting as Theorem 4.1 of AdaMem? This is to provide a setting where the convergence rate with momentum is better, and thus it could be understood how the method interpolates between the two rates.

--------------------------------

Updated the score based on rebuttal.

**Relation To Broader Scientific Literature:**

The work is well placed within the literature of memory efficient optimizers.

**Theoretical Claims:**

No I did not verify the correctness of the theoretical claims.

---

> ### Author Rebuttal · Authors · 2025-04-01
>
> We would like to thank the reviewer for their detailed comments. We appreciate their commendation of our extensive experimental evaluation and the ablation study we conducted. We address their concerns and questions below.
>
> The tables and proof that we will refer to using an apostrophe (e.g., Table 1') can be found at the anonymous link https://anonymous.4open.science/r/frugal-618F/rebuttal.pdf.
>
> >1. Would it be possible to rerun some of the pretraining experiments with beta2 = 0.95?
>
> We conducted additional experiments with a $\beta_2=0.95$ for models with sizes 60M, 130M, and 350M. The results are presented at Table 1'.
>
> The results are consistent with the observations from the main paper- FRUGAL performs significantly closer to the full-rank baseline than GaLore and BAdam.
>
> We would like to point out that, when selecting the value for $\beta_2$ for the main experiments in the paper, we followed the setup from our primary baseline, GaLore. We agree that $\beta_2=0.95$ is a standard choice for training LLMs.
>
> >3. How does the proposed method differ from AdaMem?
>
> We thank the reviewer for bringing this paper to our attention. This algorithm indeed shares similarities with our framework, as it also uses residual for ensuring a full rank update. We will certainly include a discussion of this paper in the camera-ready version.
>
> We would also like to emphasize that FRUGAL is a more general method compared to AdaMeM. Specifically, Algorithm 1 allows for choosing among various types of state-free optimizers, state-full optimizers, and projections, making it significantly more flexible than AdaMeM. Furthermore, AdaMeM can be considered a special case of our proposed FRUGAL framework, with Adafactor with momentum as the state-full optimizer, one-sided Adafactor as the state-free optimizer, and SVD-based projection.
>
> >2. Would it be possible to include AdaMem as a baseline?
>
> We reimplemented AdaMeM (see GaLore-based implementation at the https://anonymous.4open.science/r/frugal-618F/adamem.py) and conducted pre-training experiments on models up to 350M in size. Following the original work, we swept the learning rate over the range [1e-4, 3e-4, 1e-3, 3e-3] and the delta parameter over [0.5, 1.0]. The results in Table 2' indicate that AdaMeM slightly underperforms compared to the FRUGAL.
>
> >4. Can the gains of the given method be theoretically shown in a similar setting as Theorem 4.1 of AdaMem? This is to provide a setting where the convergence rate with momentum is better, and thus, it could be understood how the method interpolates between the two rates.
>
> We agree with the reviewer that having an analysis for such a setup would help provide better intuition about the capabilities of the proposed method, and the paper would definitely benefit from it. However, such an analysis is a non-trivial task, and our attempts during the process of creating a paper showed that it is not readily obvious how to achieve such a result.
>
> Regarding Theorem 4.1 from [1], we believe there was a mistake in its proof, and the theorem is *incorrect*. See the description of the error in Proof 1'.
>
> While slightly modifying the formulation—specifically, replacing top-k with bottom-k—could yield desirable results, we believe they would not be useful or illustrative. This is because applying accelerated methods to the bottom-k eigenspace is somewhat counterintuitive and contradicts methods used in practice (including AdaMeM). Furthermore, we believe that considering a quadratic setup with SVD is an oversimplified setup as computing SVD is harder than solving the problem and yields a closed-form solution, so no optimization is needed beyond SVD.
>
> We would also like to emphasize that the existing analysis presented in Section 5 is conducted under assumptions that closely match real-world conditions, achieves optimal convergence rates, and thus serves as a significant theoretical grounding for our method.
>
> We would be glad to continue the discussion and address any follow-up questions the reviewer may have. For now we believe that we addressed all the reviewer's concerns and questions, none of which is a serious issue with our approach. Therefore, we would kindly request the reviewer to reconsider their score.

---

> > ### Comment · Reviewer_wtSQ · 2025-04-03
> >
> > Thanks for the responses. I am happy to see that the results hold for $\beta_2=0.95$ as well. Also, happy to see the comparison to Adamem. However, I think the proof is correct in the AdaMem work, it's just that I believe the assumption is stated wrongly, it should be $\lambda_i \propto i^{-\alpha}$, as this is the standard power-law decay assumption, where $\lambda_1$ represents the maximum eigenvalue and $\lambda_d$ represents the minimum eigenvalue.
> >
> > Based on the results, I am happy to update my score.

---

### Official Review · Reviewer_HZzU · 2025-03-15

**Overall Recommendation:** 3

**Summary:**

The paper introduces FRUGAL, a memory-efficient optimization framework designed for scalable training of large language models (LLMs). The key idea behind FRUGAL is gradient splitting, which enables a mix of stateful optimizers (e.g., AdamW) for a low-dimensional subspace and state-free optimizers (e.g., signSGD, SGD) for the remaining directions. This allows full-rank parameter updates while keeping memory overhead low.

**Claims And Evidence:**

1. FRUGAL enables full-rank updates with lower memory overhead than existing methods. Supported by experimental results showing that FRUGAL achieves performance close to full AdamW training while using significantly less memory.

2. State-free optimizers (e.g., signSGD) are effective for certain LLM components. The authors provide evidence that embeddings, RMSNorms, and all but the Logits layer can be trained with signSGD with minimal accuracy loss.

3. FRUGAL achieves state-of-the-art memory-efficient training for both pre-training and fine-tuning. Results show that FRUGAL outperforms GaLore and BAdam in pre-training while achieving comparable fine-tuning performance to LoRA with lower memory costs.

4. FRUGAL maintains convergence rates comparable to standard optimizers, supported by theoretical proofs

**Essential References Not Discussed:**

Not I aware of

**Experimental Designs Or Analyses:**

The experimental evaluation includes:
1. Pre-training performance (LLaMA, C4 dataset)
2. Fine-tuning performance (RoBERTa, GLUE benchmark)

Findings:
1. FRUGAL achieves similar performance to AdamW at lower memory cost.
2. It outperforms GaLore and BAdam on pre-training tasks.
3. SignSGD works well for embeddings and normalization layers but degrades performance for the Logits layer.

**Methods And Evaluation Criteria:**

FRUGAL splits gradient updates into two subspaces:
1. Stateful subspace (L) – Updated using AdamW or another optimizer that maintains state.
2. State-free subspace (M) – Updated using signSGD or SGD, eliminating the need for momentum and variance buffers.

Also it supports various subspace selection strategies, including:
1. SVD-based projections (like GaLore).
2. Random projections (for computational efficiency).
3. Block-wise updates (similar to BAdam).

**Other Comments Or Suggestions:**

None

**Other Strengths And Weaknesses:**

None

**Questions For Authors:**

Computational Overhead: How does FRUGAL compare in terms of computational efficiency, compared to GaLore and BAdam?

Hyperparameter Sensitivity: most of experiments set rho=0.25. How does it get selected? How about its sensitivity?

**Relation To Broader Scientific Literature:**

memory-efficient training approaches (LoRA, GaLore, BAdam)

**Theoretical Claims:**

The paper provides theoretical guarantees for FRUGAL’s convergence. The proofs are generally correct.

---

> ### Author Rebuttal · Authors · 2025-04-01
>
> We thank the reviewer for their feedback and address their questions below.
>
> The tables and proof that we will refer to using an apostrophe (e.g., Table 1') can be found at the anonymous link https://anonymous.4open.science/r/frugal-618F/rebuttal.pdf.
>
> >Computational Overhead: How does FRUGAL compare in terms of computational efficiency, compared to GaLore and BAdam?
>
> We have measured the running time of all methods used in the paper. We present the average computational time of the optimizer step for different sizes of LLaMA models in Table 7'. The measurements for memory-efficient methods were made with density $\rho=0.25$ and update gap $T$ equal to $200$. We report the average time over 200 steps (to capture precisely one step with the state-full subspace update). Measurements were conducted on a single A100-80G GPU using PyTorch 2.4.1. We note that these experiments were conducted without using `torch.compile`.
>
> The results show that memory-efficient methods requiring gradient projection within each Linear layer matrix (GaLore, RandK) stand out negatively. GaLore requires more time than RandK due to SVD decomposition. As model size increases, blockwise-projection methods even start outperforming Adam, despite being implemented through a for-loop over all parameters, while PyTorch uses an efficient Adam implementation by stacking updates into a single shared tensor (flag `foreach=True`) to better utilize the parallelization capabilities of modern GPUs. This occurs because Adam's update step requires significantly more operations than the state-free step in FRUGAL. Therefore, approximately 75% of updates in FRUGAL's for-loop require significantly fewer computations and, consequently, less time.
>
> >Hyperparameter Sensitivity: most of experiments set rho=0.25. How does it get selected?
>
> The value $\rho=0.25$ was chosen to match the number of trainable parameters in the GaLore baseline, where the rank of the projection in most experiments was $1/4$ of the hidden size. We appreciate the reviewer's question and will add this explanation to the text.
>
> >How about its sensitivity?
>
> We conducted an ablation study to verify the robustness of our algorithm to density $\rho$. This experiment is described in Section 6.4, lines 420-422, and in Table 15. The results indicate that perplexity increases gradually as we transition from $\rho=1.0$ (which essentially coincides with AdamW) to $\rho=0.0$.

---

### Decision · Program_Chairs · 2025-05-01

**Decision:**

Accept (poster)

**Comment:**

The authors propose a hybrid optimization method aimed at improving memory efficiency without compromising performance. The core idea is intuitive and somewhat straightforward, with similar approaches having been explored in the literature. It is recommended that the authors cite them in relevant works, such as Fira [1]. Therefore, the novelty of the methodology appears incremental.

Nevertheless, the reviewers recognize the value and contributions of this work, particularly noting the strength of the empirical evaluation and comprehensive ablation studies provided. Additionally, exploring the method's performance in scenarios involving longer context lengths and over-training setups could further highlight the robustness and practical utility of the proposed approach.

[1] Chen, Xi, et al. "Fira: Can We Achieve Full-rank Training of LLMs Under Low-rank Constraint?." arXiv preprint arXiv:2410.01623 (2024).